# UNDERSTANDING OPTIMIZATION OF OPERATOR NETWORKS WITH VARIATIONAL LOSS FOR SOLVING PDES

## ABSTRACT

In this paper, we analyze the optimization of operator networks for solving elliptic PDEs with variational loss functions. While approximation and generalization errors in operator networks have been extensively studied, optimization error remains largely unexplored. We apply Restricted Strong Convexity (RSC) theory to rigorously examine the optimization dynamics of operator networks trained with variational loss, providing theoretical guarantees for convergence and training stability. We further investigate the role of the condition number of $A$ in optimization and demonstrate that preconditioning strategies significantly improve convergence rates, establishing a solid theoretical basis for the empirical benefits of preconditioning. We also address the lower bound of a key quantity, $q_t$, which ensures convergence. To prevent $q_t$ from vanishing, we propose an algorithm that adaptively incorporates additional weights into the variational loss function, leveraging values already computed during training, thereby avoiding any extra computational costs. Finally, we validate our theoretical assumptions through numerical experiments, demonstrating their practical applicability and confirming the effectiveness of preconditioning, with significant improvements in training performance and convergence rates.

## 1 INTRODUCTION

Scientific machine learning (SciML) has advanced through approaches like physics-informed neural networks (PINNs) (Raissi et al., 2019), the Deep Ritz Method (DRM) (Yu et al., 2018), and the Deep Galerkin Method (DGM) (Sirignano & Spiliopoulos, 2018), which use neural networks to approximate solutions to complex partial differential equations (PDEs). Additionally, operator learning methods, such as the Deep Operator Network (DeepONet) (Lu et al., 2021) and Fourier Neural Operator (FNO) (Li et al., 2020), map input parameters (e.g., initial/boundary conditions or forcing terms) directly to PDE solutions. Originally developed as supervised learning methods, DeepONet and FNO have later been expanded to include the principles of PINNs, enabling them to work in unsupervised learning scenarios as well. They are referred to as Physics-Informed Deep Operator Networks (PI-DeepONet) (Wang et al., 2021b) and Physics-Informed Neural Operators (PINO) (Li et al., 2024), respectively. As a result, these methods allow for the modeling of intricate physical systems while reducing dependency on extensive labeled datasets.

However, many of these methods face optimization challenges, particularly in imposing accurate boundary conditions and balancing the physics-informed loss with the boundary loss (Wong et al., 2022; Krishnapriyan et al., 2021). Furthermore, the loss functions typically involve derivatives of the network with respect to input variables, leading to a highly intricate optimization landscape. Additionally, even when the loss functions converge near zero, there is no guarantee that the approximate solutions are close to the true PDE solutions.

In recent years, operator learning methods based on the variational loss form have gained attention as a means to improve the accuracy and efficiency of solving PDEs. Notable examples include the Finite Element Operator Network (FEONet) (Lee et al., 2023) and the Unsupervised Legendre Galerkin Network (ULGNET) (Choi et al., 2023), which approximate PDE solutions by combining neural networks with classical numerical methods, such as the Finite Element Method and Spectral Methods. These methods effectively circumvent the aforementioned issues by incorporating basis functions. The use of basis functions, combined with variational losses, not only eliminates the need

for additional penalty losses to impose boundary conditions but also simplifies the loss structure by removing the need to differentiate neural networks with respect to input variables. This enables them to handle singularly perturbed problems, boundary layer problems, and complex geometries more efficiently (Choi et al., 2023; Lee et al., 2023).

With the advancement of machine learning techniques, the need for rigorous theoretical analysis has become increasingly evident. In particular, the demand for a deeper understanding of optimization, which significantly affects both the efficiency and stability of training, has grown substantially. Most traditional optimization frameworks are primarily grounded in convex settings (Boyd & Vandenberghe, 2004). However, it is well known that deep learning models are challenging to analyze within these frameworks (Liu et al., 2020). To address this limitation, the Neural Tangent Kernel (NTK) theory (Jacot et al., 2018), along with the PŁ-condition (Garrigos & Gower, 2023), has been developed to provide convergence guarantees, drawing significant attention in the deep learning community (Du et al., 2019; Allen-Zhu et al., 2019; Zou et al., 2020; Wang et al., 2022; 2021a; Gao et al., 2023). Nonetheless, these theories often necessitate that the model be infinitely wide or confined to the near-initialization regime, which considerably diverges from practical applications. Another noteworthy framework for analyzing optimization is the Restricted Strong Convexity (RSC) framework. While RSC has been extensively studied in linear or convex settings (Wainwright, 2019; Negahban & Wainwright, 2012; Zhang & Cheng, 2015), its application to deep learning models has only been explored in recent years. Motivated by this, we aim to extend the RSC theory to SciML techniques, which are known to be more challenging to analyze than traditional machine learning methods. Specifically, our main contributions are as follows:

- We apply the RSC theory to investigate the optimization process using GD in unsupervised operator learning methods with variational loss. Our analysis rigorously proves the convergence of the optimization error, providing theoretical guarantees for training stability.

- Building on these theoretical insights, we examine the impact of the condition number of $\boldsymbol{A}$, which is determined by the PDE structures in operator networks using variational loss forms, on the optimization process. We demonstrate how preconditioning strategies can significantly improve convergence rates. This provides a solid theoretical foundation for the empirical observation that preconditioning enhances training efficiency.

- In addition, we propose an algorithm that adaptively improves the lower bound of $q_t$, a key quantity in optimization dynamics that ensures convergence. By adjusting the weights in the variational loss function, the proposed algorithm prevents $q_t$ from vanishing during the training process, leading to improved convergence rates.

- Through numerical experiments, we validate the assumptions underlying our theoretical analysis, demonstrating that they hold in most practical cases. These experiments further confirm the effectiveness of preconditioning, showing significant improvements in both training performance and convergence rates.

## 2 PRELIMINARY

We begin with a brief overview of neural networks, followed by an introduction to the basic concepts of operator networks based on variational loss, which are the primary focus of our optimization theory.

### 2.1 NEURAL NETWORKS

For any inputs feature $\omega \in \Omega$, let us define $\omega := \alpha^{(0)}(\omega)$ and $m_0 := M$ (to be determined later) for a convenience. Then, a fully-connected neural network is defined by the following recursion relation:

$$\alpha^{(l)} = \phi\left(\frac{1}{\sqrt{m_{l-1}}} \boldsymbol{W}^{(l)} \alpha^{(l-1)}\right), \quad \boldsymbol{W}^{(l)} \in \mathbb{R}^{m_l \times m_{l-1}}, \ l \in [L],$$

$$\widehat{\alpha}(\omega) = \alpha^{(L+1)}(\omega) = \frac{1}{\sqrt{m_L}} \boldsymbol{V}^T \alpha^{(L)}(\omega), \quad \boldsymbol{V} \in \mathbb{R}^{N \times m_L},$$

where $m_l$ is the width of the $l$-th layer, $L$ is the depth of the network, $\phi$ is the activation function, $\boldsymbol{W}^{(l)}$ is the weight matrix, and $\boldsymbol{V}$ is the weight vector for the last layer. Throughout this paper, for

simplicity, for simplicity, we assume that the width $m_l$ ($l = 1, \cdots, L$) of all the layer is the same as $m$. Then, we denote the set of all parameters of the network by $\theta$ as follows:

$$\theta = \text{vec}(\boldsymbol{W}^{(0)}, \boldsymbol{W}^{(1)}, \ldots, \boldsymbol{W}^{(L)}, \boldsymbol{V}) \in \mathbb{R}^p,$$

where $p = m(M + m^{L-1} + N)$. The model parameters $\theta$ are updated during optimization processes for minimizing a suitably given loss function.

## 2.2 OPERATOR NETWORK WITH THE VARIATIONAL LOSS FORM

In this subsection, we briefly introduce the operator network with the variational loss, focusing on the specific examples. For simplicity, we will consider self-adjoint second ordered Elliptic PDEs with homogeneous Dirichlet boundary condition on the compact domain $D \subset \mathbb{R}^d$:

$$\begin{aligned} -\text{div}(\boldsymbol{a}(\boldsymbol{x})\nabla u(\boldsymbol{x})) + \boldsymbol{b}(\boldsymbol{x}) \cdot \nabla u(\boldsymbol{x}) + c(\boldsymbol{x})u &= g(\boldsymbol{x}) \text{ in } D, \\ u(\boldsymbol{x}) &= h(\boldsymbol{x}) \text{ on } \partial D, \end{aligned} \tag{1}$$

where the coefficient $\boldsymbol{a}(\boldsymbol{x})$ is uniformly elliptic and $c(x) \geq 0$. The weak solution of equation 33 is defined by the function $u(\boldsymbol{x})$ satisfying the following variational formulation:

$$\int_D \boldsymbol{a}(\boldsymbol{x})\nabla u(\boldsymbol{x}) \cdot \nabla v(\boldsymbol{x}) + [\boldsymbol{b}(\boldsymbol{x}) \cdot \nabla u(\boldsymbol{x}) + c(\boldsymbol{x})u(\boldsymbol{x})]\, v(\boldsymbol{x})d\boldsymbol{x} = \int_D g(\boldsymbol{x})v(\boldsymbol{x})d\boldsymbol{x}, \quad \forall v \in H_0^1(D). \tag{2}$$

We note that the existence and uniqueness of weak solutions is obtained by the Lax-Milgram theorem (Brenner, 2008). Traditional numerical methods such as the Finite Element Method (FEM) and Spectral Method approximate the weak solution $u(\boldsymbol{x})$ by a linear combination of basis functions $\phi_k(\boldsymbol{x})$. Specifically, the solution $u(\boldsymbol{x})$ is approximated by $u_N(\boldsymbol{x}) = \sum_{k=1}^N \alpha_k \phi_k(\boldsymbol{x})$, where $\{\phi_k(\boldsymbol{x})\}_{k=1}^N$ are chosen basis functions, typically selected based on the geometry of the domain $D$, the boundary condition of the PDE, and the other properties of the PDE. For FEM, these basis functions are piecewise polynomials defined over a mesh that discretizes the domain, while Spectral Methods use globally defined functions such as trigonometric functions or Legendre-Galerkin polynomials. In both approaches, the coefficients $\alpha_k$ can be determined by solving the discrete approximation of the variation formulation equation 2 with given basis functions $\phi_k(\boldsymbol{x})$:

$$\int_D \boldsymbol{a}(\boldsymbol{x})\nabla u_N(\boldsymbol{x}) \cdot \nabla \phi_k(\boldsymbol{x}) + [\boldsymbol{b}(\boldsymbol{x}) \cdot \nabla u_N(\boldsymbol{x}) + c(\boldsymbol{x})u_N(\boldsymbol{x})]\, \phi_k(\boldsymbol{x})d\boldsymbol{x} = \int_D g(\boldsymbol{x})\phi_k(\boldsymbol{x})d\boldsymbol{x}, \quad \forall k. \tag{3}$$

which can be rewritten as the linear algebraic system,

$$\boldsymbol{A}\boldsymbol{\alpha} = \boldsymbol{g}, \tag{4}$$

where $\boldsymbol{\alpha} := (\alpha_1, \cdots, \alpha_N)^\top$, and $\boldsymbol{A}$ and $\boldsymbol{g}$ are given as follows:

$$A_{ij} = \int_D \boldsymbol{a}(\boldsymbol{x})\nabla \phi_i \cdot \nabla \phi_j + [\boldsymbol{b}(\boldsymbol{x}) \cdot \nabla \phi_i(\boldsymbol{x}) + c(\boldsymbol{x})\phi_i(\boldsymbol{x})]\, \phi_j(\boldsymbol{x})d\boldsymbol{x}, \; g_j = \int_D g(\boldsymbol{x})\phi_j d\boldsymbol{x}.$$

Now, we are ready to introduce the method we are mainly concern with in this paper. The operator network with the variational loss form involving Unsupervised Legendre-Galerkin neural network (ULGNET) (Choi et al., 2023) and Finite Element Operator Network (FEONet) (Lee et al., 2023) are based on the aforementioned classical numerical methods with finite basis functions. For each PDEs, these methods approximate the coefficient $\alpha_k$ as an output of a neural network with variable coefficients and forcing term as an input, instead of solving the linear algebraic equation equation 4. For clarify of presentation, we only consider that an input for the neural network is the forcing term $g(\boldsymbol{x})$. Let the forcing term $\boldsymbol{g}$ be parametrized by the random parameter $\omega$ in the compact parameter space $\Omega$. For each $g(\boldsymbol{x}, \omega)$, the coefficients $\widehat{\alpha}_k(\omega; \theta)$ are generated as an output of the neural network and the solution is approximated by $\widehat{u}_N(\boldsymbol{x}, \omega; \theta) = \sum_{k=1}^N \widehat{\alpha}_k(\omega; \theta)\phi_k(\boldsymbol{x})$.

To train the neural network, we use the following variation loss, inspired by equation 3:

$$\mathcal{L}(\theta) = \mathbb{E}_{\omega \sim \Omega}\bigg[\sum_{k=1}^N \bigg| \int_D \bigg\{ \boldsymbol{a}(\boldsymbol{x})\nabla \widehat{u}_N(\boldsymbol{x}, \omega; \theta) \cdot \nabla \phi_k(\boldsymbol{x})$$

$$+ [b(\boldsymbol{x}) \cdot \nabla u_N(\boldsymbol{x}, \omega; \theta) + c(\boldsymbol{x})\widehat{u}_N(\boldsymbol{x}, \omega; \theta)]\, \phi_k(\boldsymbol{x}) \bigg\}d\boldsymbol{x} - \int_D g(\boldsymbol{x}, \omega)\phi_k(\boldsymbol{x})d\boldsymbol{x} \bigg|^2 \bigg]. \tag{5}$$

In practice, for the computational efficiency, we can deal with the empirical variational loss instead of equation 5:

$$\mathcal{L}^M(\theta) = \frac{|\Omega|}{M} \sum_{j=1}^{M} \sum_{k=1}^{N} \left| \int_D \left\{ \boldsymbol{a}(\boldsymbol{x}) \nabla \widehat{u}_N(\boldsymbol{x}, \omega_j; \theta) \cdot \nabla \phi_k(\boldsymbol{x}) \right. \right.$$

$$\left. \left. + \left[ \boldsymbol{b}(\boldsymbol{x}) \cdot \nabla u_N(\boldsymbol{x}, \omega_j; \theta) + c(\boldsymbol{x}) \widehat{u}_N(\boldsymbol{x}, \omega_j; \theta) \right] \phi_k(\boldsymbol{x}) \right\} d\boldsymbol{x} - \int_D g(\boldsymbol{x}, \omega_j) \phi_k(\boldsymbol{x}) d\boldsymbol{x} \right|^2,$$

where $M$ is the number of training samples. This empirical loss can be written as the vectorized form:

$$\mathcal{L}^M(\theta) = \frac{|\Omega|}{M} \sum_{j=1}^{M} \|\boldsymbol{A}\widehat{\alpha}(\omega_j) - \boldsymbol{g}_j\|_2^2, \tag{6}$$

where $\widehat{\alpha} = (\widehat{\alpha}_1, \cdots, \widehat{\alpha}_N)^\top \in \mathbb{R}^N$ is an approximate coefficient vector obtained by the neural network, and $\boldsymbol{A}$ and $\boldsymbol{g}$ are given as in equation 4. Throughout the remainder part of this paper, we will use the above vectorized formulation.

Based on the classical numerical theory, the basis functions $\phi_k(\boldsymbol{x})$ are chosen in a way that the approximate solution $\widehat{u}_N$ directly satisfies the boundary conditions regardless of the choice of coefficient $\widehat{\alpha}_k$. Therefore, unlike other unsupervised operator learning frameworks that typically require additional loss terms to enforce boundary conditions, our approach inherently satisfies boundary conditions without an extra penalty loss. Moreover, all differential operators with respect to the input variable $\boldsymbol{x}$ are applied to the fixed basis functions $\phi_k(\boldsymbol{x})$ in the loss $\mathcal{L}^M$. This intrinsic structure of operator learning methods with the variational loss results in that the loss $\mathcal{L}^M$ consists of a single variational-type term and does not involve any derivatives of the neural network with respect to $\boldsymbol{x}$. Consequently, complex unsupervised learning tasks for solving PDEs are transformed into simpler tasks like data-fitting supervised learning with a standard least squares loss.

For more details and performance in various numerical tests, we refer to Choi et al. (2023); Lee et al. (2023). Mathematical studies for operator learning methods with the variational loss form can be found in (Hong et al., 2024; Ko et al., 2022).

## 3   RESTRICTED STRONG CONVEXITY

In this section, we analyze the optimization process of operator learning methods with variational loss using RSC theory. This provides an alternative convergence theory to the commonly used NTK-based approaches. From this analysis, we explore the impact of the condition number on convergence and also discuss the relationship between RSC theory and NTK theory.

### 3.1   RSC THEORY IN THE OPERATOR NETWORK WITH THE VARIATIONAL LOSS FORM

In this section, we develop a theoretical framework for analyzing the optimization process of operator learning methods based on the variational form using RSC theory. However, it is important to note that this analysis is not restricted to our operator learning method alone. The same framework can be extended to other operator learning methods or PINNs that employ variational loss functions.

RSC theory has been extensively studied in other settings, such as linear models and convex loss functions (Wainwright, 2019; Negahban & Wainwright, 2012; Zhang & Cheng, 2015). More recently, in the work of Banerjee et al. (2022), RSC theory has been applied to analyze the optimization process of deep learning models for simple supervised learning tasks. Naturally, one might hope to extend this analytical framework to the training dynamics of scientific machine learning (SciML) methods. However, for many unsupervised approaches that use PDE residual loss functions, intended to embed the physical laws described by PDEs into neural networks, the inherent complexity of these loss functions poses significant challenges for directly applying optimization theories, including RSC.

In contrast, as mentioned in Subsection 2.2, the methods of operator learning based on the variation loss effectively circumvent these issues by leveraging the variational form of PDEs and employing basis functions. Building on this advantage, we extend the RSC-based analysis to operator

learning methods based on the variational form, specifically examining how the convergence rate is influenced by the condition number of the matrix $\boldsymbol{A}$. The condition number $\kappa(\boldsymbol{A})$ is defined as $\kappa(\boldsymbol{A}) = \sigma_{\max}(\boldsymbol{A})/\sigma_{\min}(\boldsymbol{A})$ where $\sigma_{\max}(\boldsymbol{A})$ and $\sigma_{\min}(\boldsymbol{A})$ represent the largest and smallest singular values, respectively.

In this paper, we focus mainly on analyzing the optimization of operator learning methods based on the variational form in relation to the condition number of $\boldsymbol{A}$, following the original work established in Banerjee et al. (2022). Let us begin with providing the following standard assumptions as used in Banerjee et al. (2022).

**Assumption 1** (Activation). *The activation $\phi$ is 1-Lipschitz i.e., $|\phi'| \leq 1$, and $\beta_\sigma$ smooth, i.e., $|\sigma''| \leq \beta_\phi$.*

**Assumption 2** (Weight initialization). *For $l \in [L]$, the weights are initialized as $\mathrm{W}_{0,ij}^{(l)}$, $\mathrm{V}_{0,ij} \sim \mathcal{N}(0, \sigma_0^2)$, where $\sigma_0 = \frac{\sigma_1}{2(1+\frac{2\sqrt{\log m}}{\sqrt{m}})}$, $\sigma_1 > 0$, and $\boldsymbol{V}_0$ is a random unit matrix, i.e. $\|\boldsymbol{V}_0\|_2 = 1$.*

**Assumption 3** (Boundedness of input features). *For every $\omega \in \Omega$, there exists a $M > 0$ such that $\|\omega\|_2^2 \leq M$.*

We also provide the following definition.

**Definition 3.1.** *Given a set of parameters $\bar{\theta} \in \mathbb{R}^m$, we define three subsets in the space of the model parameters as follows:*

$$B_{\rho,\rho_1}^{Spec}(\bar{\theta}) := \left\{ \theta \in \mathbb{R}^p | \ \|\boldsymbol{W}^{(l)} - \bar{\boldsymbol{W}}^{(l)}\|_2 \leq \rho, l \in [L], \|\boldsymbol{V} - \bar{\boldsymbol{V}}\|_2 \leq \rho_1 \right\}, \tag{7}$$

$$B_\rho^{Euc}(\bar{\theta}) := \left\{ \theta \in \mathbb{R}^p | \ \|\theta - \bar{\theta}\|_2 \leq \rho \right\}, \tag{8}$$

$$Q_q(\bar{\theta}) := \left\{ \theta \in \mathbb{R}^p | \ \frac{1}{M} \sum_{i=1}^M \|\nabla_\theta \widehat{\alpha}(\omega_i; \theta)(\theta - \bar{\theta})\|_2^2 > q\|\theta - \bar{\theta}\|_2^2 \right\}, \tag{9}$$

*where $\bar{g}$ represents any matrix having a suitable column dimension to be multiplicable with $\theta$, and $\|\cdot\|$ denotes the spectral norm for matrices while denoting the $L_2$ norm for vectors.*

For simplicity, we deal with the model parameter $\theta$ only in the ball $B_{\rho,\rho_1}^{\mathrm{Spec}}(\theta_0)$ for some given initial parameter $\theta_0$. Here, the radius $\rho$ is chosen in a way that $\rho < \sqrt{m}$ and consequently the constant $\sigma_1$ in Assumption 2 is fixed as $1 - \frac{\rho}{\sqrt{m}}$, which is a reasonable choice in practice. More discussion of the choice for these parameters can be found in Banerjee et al. (2022).

To establish the main theorem, we present key lemmas. In particular, we observe that the constants associated with the restricted strong convexity and smoothness of the loss function are related to $\sigma_{\min}(\boldsymbol{A})$ and $\sigma_{\max}(\boldsymbol{A})$. This observation is crucial for understanding how the condition number influences the optimization dynamics.

**Lemma 3.2.** *Consider a fixed $\theta \in B_{\rho,\rho_1}^{Spec}(\theta_0)$ and $q$ be a fixed positive constant. Under Assumptions 1, 2 and 3, the following inequality holds with probability at least $(1 - \frac{2(L+1)}{m})$: for all $\theta' \in Q_q(\theta) \cap B_{\rho,\rho_1}^{Spec}(\theta_0) \cap B_{\rho_2}^{Euc}(\theta)$*

$$\mathcal{L}^M(\theta') \geq \mathcal{L}^M(\theta) + \langle \theta' - \theta, \nabla_\theta \mathcal{L}^M(\theta) \rangle + \frac{\beta}{2}\|\theta' - \theta\|_2^2,$$

*where $\beta = (\sigma_{\min}(\boldsymbol{A}))^2 \left( q - \frac{2\varrho c_H N \rho_2}{\sqrt{m}} \right) - \sigma_{\max}(\boldsymbol{A})^2 \frac{c_H \sqrt{2Nc^*}}{\sqrt{m}}$ and $\varrho$, $c_H$, and $c^*$ are given in Appendix B.1.*

*Proof.* The detailed proof is in Appendix B.1. $\qquad\square$

**Lemma 3.3.** *Under Assumptions 1, 2 and 3, and with probability at least $(1 - \frac{2(L+1)}{m})$, we have that for all $\theta', \theta \in B_{\rho,\rho_1}^{Spec}(\theta_0)$,*

$$\mathcal{L}^M(\theta') \leq \mathcal{L}^M(\theta) + \langle \theta' - \theta, \nabla_\theta \mathcal{L}^M(\theta) \rangle + \frac{\gamma}{2}\|\theta' - \theta\|_2^2,$$

*where $\gamma = (\sigma_{\max}(\boldsymbol{A}))^2 \left( \varrho^2 N + \frac{c_H \sqrt{2c^* N}}{\sqrt{m}} \right)$, and $\varrho$, $c_H$, and $c^*$ are given in Appendix B.1.*

*Proof.* The detailed proof is in Appendix B.1. □

With the detailed results from Lemmas 3.2 and 3.3, we are now prepared to state the main theorem, which relates the convergence rate of the optimization process of $\mathcal{L}^M(\theta)$ in $B_{\rho,\rho_1}^{\text{Spec}}$ to the condition number $\kappa(\boldsymbol{A})$.

**Theorem 3.4** (Optimization of the variation loss). *Let $\{\theta_t\}$ denote the sequence of model parameters generated by GD with the stepsize $\eta_t = \frac{\omega_t}{\gamma} \leq \frac{2}{\gamma}$, and we define*

$$q_t = \frac{\sum_{i=1}^M \|\nabla_\theta \widehat{\alpha}(\omega_i; \theta_t) \nabla_\theta \mathcal{L}^M(\theta_t)\|_2^2}{M \|\nabla_\theta \mathcal{L}^M(\theta_t)\|_2^2}, \qquad B_t := Q_{q_t}(\theta_t) \cap B_{\rho,\rho_1}^{Spec}(\theta_0) \cap B_{\rho_2}^{Euc}(\theta_t),$$

$$\theta^* \in arginf_{\theta \in B_{\rho,\rho_1}^{Spec}(\theta_0)} \mathcal{L}(\theta), \;\; \bar{\theta}_{t+1} \in arginf_{\theta \in B_t} \mathcal{L}(\theta) \;\; and \;\; \delta_t := \frac{\mathcal{L}(\bar{\theta}_{t+1}) - \mathcal{L}(\theta^*)}{\mathcal{L}(\theta_t) - \mathcal{L}(\theta^*)}.$$

*Under Assumptions 1, 2 and 3, we further assume that for each iteration $t$, the followings holds:*

$$\text{(A1)} \quad \theta_{t+1} \in B_{\rho,\rho_1}^{Spec}(\theta_0) \cap B_{\rho_2}^{Euc}(\theta_t), \; and \qquad \text{(A2)} \qquad q_t > \frac{2\varrho c_H N \rho_2}{\sqrt{m}} + \kappa(A)^2 \frac{c_H \sqrt{2Nc^*}}{\sqrt{m}}. \tag{10}$$

*Then, we have $\delta_t \in [0, 1)$, and the following inequality holds with probability at least $(1 - \frac{2(L+1)}{m})$:*

$$\mathcal{L}^M(\theta_{t+1}) - \mathcal{L}^M(\theta^*) \leq (1 - r_t \omega_t (2 - \omega_t)(1 - \delta_t)) \left(\mathcal{L}^M(\theta_t) - \mathcal{L}^M(\theta^*)\right)$$

*where $r_t$ is given by*

$$r_t = \frac{(\kappa(A))^{-2} \left(q_t - 2\varrho N \frac{c_H \rho_2}{\sqrt{m}}\right) - \frac{c_H \sqrt{2Nc^*}}{\sqrt{m}}}{\varrho^2 N + \frac{c_H \sqrt{2Nc^*}}{\sqrt{m}}} > 0,$$

*and $\varrho$, $c_H$, and $c^*$ are given as in Lemma 3.2 and 3.3.*

*Proof.* The detailed proof is in Appendix B.2. □

## 3.2 THE IMPACT OF THE CONDITION NUMBER ON CONVERGENCE

In our RSC-based analysis of the optimization process, we established a clear connection between the convergence rate and the condition number of the matrix $\boldsymbol{A}$. Specifically, in Theorem 3.4, the quantity $r_t$, which is closely related to the convergence rate, is given as

$$r_t = \frac{(\kappa(A))^{-2} \left(q_t - 2\varrho N \frac{c_H \rho_2}{\sqrt{m}}\right) - \frac{c_H \sqrt{2Nc^*}}{\sqrt{m}}}{\varrho^2 N + \frac{c_H \sqrt{2Nc^*}}{\sqrt{m}}}.$$

This implies that a smaller condition number $\kappa(\boldsymbol{A})$ results in a faster convergence rate, which is essential for efficient training. Therefore, reducing the condition number is a crucial strategy for enhancing optimization efficiency. This finding is consistent with previous studies showing that reducing the condition number also positively impacts generalization and approximation errors (Hong et al., 2024). Thus, improving the condition number benefits optimization, generalization, and approximation errors simultaneously, without any trade-offs. These insights underscore the importance of employing training strategies that reduce the condition number. Furthermore, the assumption (A2) in Theorem 3.4 becomes less stringent when the condition number of $\boldsymbol{A}$ is reduced. Specifically, the right-hand side of (A2),

$$\frac{2\varrho c_H N \rho_2}{\sqrt{m}} + \kappa(\boldsymbol{A})^2 \frac{c_H \sqrt{2Nc^*}}{\sqrt{m}}$$

decreases as the condition number $\kappa(\boldsymbol{A})^2$ becomes smaller. Thus, a smaller condition number allows for a more relaxed bound on $q_t$, making the assumptions more applicable in practical scenarios. This further underscores the importance of reducing the condition number.

One well-known approach to reducing the condition number is the use of preconditioning. In this method, a preconditioner matrix $\boldsymbol{P}$ is applied to transform the system of equations into an equivalent system with a lower condition number. Specifically, preconditioning replaces the matrix $\boldsymbol{A}$ with $\boldsymbol{PA}$, where $\boldsymbol{P}$ is chosen to ensure that the condition number $\kappa(\boldsymbol{PA})$ is significantly smaller than that of $\boldsymbol{A}$. This transformation results in a system that is easier to optimize and converges more quickly.

By showing that a reduced condition number directly improves the convergence rate in the optimization process, our analysis explains why preconditioning leads to faster and more efficient training. This alignment between theoretical and empirical results highlights the importance of preconditioning as a key strategy in operator learning with the variational losses, particularly in FEONet and ULGNET. The numerical tests supporting these observations can be found in Section **??**, where we demonstrate the impact of preconditioning on training stability and efficiency

### 3.3 RELATION BETWEEN RSC AND NTK

In the proof of the main theorem, two technical assumptions, (A1) and (A2), are crucial for establishing the convergence of the loss $\mathcal{L}^M$. In particular, assumption (A2) plays a significant role in ensuring the restricted strong convexity of $\mathcal{L}^M$. For assumption (A2), it is evident that the value on the right-hand side decreases as the network width $m$ increases. This implies that assumption (A2) becomes less restrictive for networks with larger widths. Indeed, if $q_t$ maintains a uniform lower bound throughout the training process, we can guarantee that assumption (A2) holds for sufficiently large widths.

Interestingly, the value of $q_t$ is closely related to the Neural Tangent Kernel (NTK), which has been one of the most widely used tools for analyzing the optimization dynamics of deep learning models (Jacot et al., 2018; Du et al., 2019). More precisely, let $\nabla_\theta \boldsymbol{\alpha} \in \mathbb{R}^{NM \times p}$ be defined as

$$\nabla_\theta \boldsymbol{\alpha} := \begin{pmatrix} \nabla_\theta \widehat{\alpha}_1(\omega_1) & \nabla_\theta \widehat{\alpha}_2(\omega_1) & \cdots & \nabla_\theta \widehat{\alpha}_N(\omega_1) & \nabla_\theta \widehat{\alpha}_1(\omega_2) & \cdots & \nabla_\theta \widehat{\alpha}_N(\omega_M) \end{pmatrix}^\top,$$

which is the matrix that lists each $1 \times p$ matrix $\nabla_\theta \widehat{\alpha}_i(\omega_j)$ as a block component in column order. Then, NTK $\boldsymbol{K}(\theta)$ can be expressed as $\nabla_\theta \boldsymbol{\alpha}(\theta)^\top \nabla_\theta \boldsymbol{\alpha}(\theta)$. Note that the quantity $q_t$ can be expressed in terms of the Neural Tangent Kernel $\boldsymbol{K}(\theta_t)$.

**Theorem 3.5** (Relation between $q_t$ and NTK). *Let $\mathbb{A} \in \mathbb{R}^{NM \times NM}$ and $\boldsymbol{r} \in \mathbb{R}^{NM \times 1}$ are given by*

$$\mathbb{A} = \begin{pmatrix} \boldsymbol{A} & \boldsymbol{0} & \cdots & \boldsymbol{0} \\ \boldsymbol{0} & \boldsymbol{A} & \cdots & \boldsymbol{0} \\ \vdots & \vdots & \ddots & \vdots \\ \boldsymbol{0} & \boldsymbol{0} & \cdots & \boldsymbol{A} \end{pmatrix}, \quad \text{and} \quad \boldsymbol{r} = \begin{pmatrix} \boldsymbol{A}\widehat{\alpha}(\omega_1) - g_1 \\ \boldsymbol{A}\widehat{\alpha}(\omega_2) - g_2 \\ \vdots \\ \boldsymbol{A}\widehat{\alpha}(\omega_M) - g_M \end{pmatrix}.$$

*Then, the following relation always holds:*

$$q_t := \frac{\sum_{i=1}^M \|\nabla_\theta \widehat{\alpha}(\omega_i; \theta_t) \nabla_\theta \mathcal{L}^M(\theta_t)\|_2^2}{M \|\nabla_\theta \mathcal{L}^M(\theta_t)\|_2^2} = \frac{\|\boldsymbol{K}(\theta_t)\mathbb{A}^\top \boldsymbol{r}(\theta_t)\|_2^2}{M \boldsymbol{r}(\theta_t)^\top \mathbb{A}\boldsymbol{K}(\theta_t)\mathbb{A}^\top \boldsymbol{r}(\theta_t)}. \quad (11)$$

*Consequently, we have the following upper and lower bounds for $q_t$:*

$$\frac{\lambda_{min}(\boldsymbol{K}(\theta_t))^2}{\lambda_{max}(\boldsymbol{K}(\theta_t))} \leq q_t \leq \frac{\lambda_{max}(\boldsymbol{K}(\theta_t))^2}{\lambda_{min}(\boldsymbol{K}(\theta_t))},$$

*where $\lambda_{\min}(\cdot)$ and $\lambda_{\max}(\cdot)$ represent the minimun and maximum eigenvalues of the corresponding matrix, respectively, and the upper bound are valid only when $\lambda_{min}(\boldsymbol{K}(\theta_t)) > 0$.*

*Proof.* The detailed proof is provided in Appendix B.3. □

An important insight from this observation is that if $\sigma_{\min}(\boldsymbol{K}(\theta_t))$ and $\sigma_{\max}(\boldsymbol{K}(\theta_t))$ have uniform lower and upper bounds, respectively, then a uniform lower bound for $q_t$ is naturally guaranteed as $\Omega\left(\frac{1}{M}\right)$. This directly implies that if $m = \Omega\left((NM)^2\right)$, the positivity of $q_t$ is ensured, making the convergence process more stable and predictable. Thus, maintaining tightly controlled singular values of $\boldsymbol{K}(\theta_t)$ contributes directly to improving the efficiency and robustness of the training process.

Although the existence of uniform positive lower and upper bounds for all eigenvalues of the NTK provides a sufficient condition to ensure that assumption (A2) holds, numerous studies have shown that guaranteeing this bound can be challenging in practice. Moreover, adaptively controlling the NTK at each training step by calculating its eigenvalues or approximations introduces significant computational overhead. To address these challenges, in the next section, we propose an algorithm that enhances the stability of the lower bound of $q_t$ by utilizing only values already computed during training, thereby avoiding the need for costly NTK calculations.

## 4   AN ADAPTIVE WEIGHT ALGORITHM

As discussed in Section 3.3, ensuring a uniform lower bound on $q_t$ is crucial to establish geometric convergence as in Theorem 3.4. However, in general, our numerical experiments indicate that the behavior of $q_t$ can often be highly unpredictable, which can be found in C.5. This unpredictability in the dynamics of $q_t$ can ultimately result in convergence failures. Therefore, controlling the behavior of $q_t$ is crucial to ensuring successful convergence.

To address this challenge, we propose an algorithm that adaptively applies weights to the loss function. This approach establishes a new lower bound for $q_t$, which is both simpler and more practical compared to the NTK-related bound discussed in Subsection 3.3. Furthermore, our experiments confirm that this strategy makes the behavior of $q_t$ more stable compared to when the strategy is not applied, throughout the entire training process, further demonstrating its effectiveness.

The remainder of this section is organized as follows. First, we establish some notations to clearly present the algorithm and underlying concepts. Next, we introduce the algorithm designed to enhance the performance of operator learning with variational losses. Finally, we provide a brief overview of the key idea behind the proposed algorithm. For clarity, we define some notations. Recalling the notation $r$ from Section 3.3, we rewrite the quantity $q_t$ and the variational loss $\mathcal{L}^M$ as follows:

$$q_t = \frac{1}{M} \sum_{j=1}^{M} \frac{\|\nabla_\theta \widehat{\alpha}(\omega_i; \theta_t) \nabla_\theta \boldsymbol{\alpha}(\theta_t) \mathbb{A}^\top \boldsymbol{r}(\theta_t)\|_2^2}{\|\nabla_\theta \boldsymbol{\alpha}(\theta_t) \mathbb{A}^\top \boldsymbol{r}(\theta_t)\|_2^2}, \text{ and } \quad \mathcal{L}^M(\theta) = \frac{1}{2M} \boldsymbol{r}^\top \boldsymbol{r}.$$

Note that multiplying both sides of Equation (6) by a non-singular matrix $\Lambda$, $\Lambda \boldsymbol{A}\widehat{\alpha}(\omega_i) = \Lambda \boldsymbol{g}_i$, $i = 1, 2, \ldots, M$, yields the same solution as the original equation (6). We refer to such a matrix $\Lambda$ as the weight matrix. In our algorithm, the weight matrix is block diagonal and varies at each time step $t$, denoted as $\Lambda_t$. Each block component of $\Lambda_t$ is represented as $\Lambda_{i,t}$ for $i = 1, \ldots, M$, i.e., $\Lambda_t = \text{Diag}(\Lambda_{1,t}, \ldots, \Lambda_{M,t})$. Furthermore, $\Lambda_{i,t}$ is a diagonal matrix having $\lambda_{ij,t}$ as the $j$-th diagonal component, multiplied by $(\boldsymbol{A}^\top)^{-1}$, i.e. $\Lambda_{i,t} = Diag(\lambda_{i1,t}, \lambda_{i2,t}, \ldots, \lambda_{iN,t})(\boldsymbol{A}^\top)^{-1}$. Using the weight matrix $\Lambda_t$, we define a modified loss function in place of the original loss $\mathcal{L}^M$ such that $\tilde{\mathcal{L}}_t^M(\theta) := \frac{1}{2M} \tilde{\boldsymbol{r}}_t^\top \tilde{\boldsymbol{r}}_t$, where $\tilde{\boldsymbol{r}}_t := \Lambda \boldsymbol{r}_t$. Here, $\tilde{r}_{ij,t} = (\boldsymbol{A}\widehat{\alpha}(\omega_i; \theta) - \boldsymbol{g}_i)_j \lambda_{ij,t}$.

With the above notation established, we present the adaptive weight algorithm as follows:

---
**Algorithm 1** Gradient Descent with Adaptive weight strategy.

---
   **Input** $\mathbb{A}$, $\boldsymbol{r}_t$, $\tilde{\boldsymbol{r}}_{t-1}$

When $\Lambda_0 := \frac{1}{\sqrt{NM}} \text{diag}(\boldsymbol{1}_{NM})$

**Require:** $N \geq 1$, $M > 1$

---
   $\lambda_{ij,t} \leftarrow \sqrt{\frac{1}{NM-1}(1 - \frac{\tilde{r}_{ij,t-1}^2}{\|\tilde{\boldsymbol{r}}_{t-1}\|^2})}$

---
   **Output** $\tilde{\mathcal{L}}_t^M, \tilde{\boldsymbol{r}}_t$

---
We provide a brief outline of the above algorithm. Let us examine the modified $q_t$ resulting from the adaptive weight methods such that

$$q_t = \frac{1}{M} \sum_{j=1}^{M} \frac{\|\nabla_\theta \widehat{\alpha}(\omega_i; \theta_t) \nabla_\theta \boldsymbol{\alpha}(\theta_t) \boldsymbol{\Lambda}_t \mathbb{A}^\top \tilde{\boldsymbol{r}}_t\|_2^2}{\|\nabla_\theta \boldsymbol{\alpha}(\theta_t) \boldsymbol{\Lambda}_t \mathbb{A}^\top \tilde{\boldsymbol{r}}_t\|_2^2}. \tag{12}$$

As we mentioned above, it is often observed that $q_t$ decreases as the iterations progress. This means that $\boldsymbol{\Lambda}_t \mathbb{A}^\top \tilde{\boldsymbol{r}}_t$ gradually becomes nearly orthogonal to $\nabla_\theta \widehat{\alpha}(\omega_i; \theta_t)$ during the training process. In this regard, our main idea is to ensure that $\boldsymbol{\Lambda}_t \mathbb{A}^\top \tilde{\boldsymbol{r}}_t$ remains parallel to the initial direction $\boldsymbol{r}_0$,

while also avoiding placement in the null space of $\nabla_\theta \widehat{\alpha}(\omega_i; \theta_t)$ throughout the training process by adaptively selecting the appropriate weight matrix $\Lambda_t$.

Let us consider $\boldsymbol{\Lambda}_t \mathbb{A}^\top \tilde{\boldsymbol{r}}_t$. Each $ij$ component of $\boldsymbol{\Lambda}_t \mathbb{A}^\top \tilde{\boldsymbol{r}}_t$ can be expressed as $(\boldsymbol{\Lambda}_t \mathbb{A}^\top \tilde{\boldsymbol{r}}_t)_{ij} = \tilde{r}_{ij,t} \lambda_{ij,t}$. By appropriately selecting $\lambda_{ij,t}$, we aim to keep each $r_{ij,t} \lambda_{ij,t}$ constant throughout the training process, thereby ensuring that the direction of $\boldsymbol{\Lambda}_t \mathbb{A}^\top \tilde{\boldsymbol{r}}_t$ remains unchanged. In fact, when applying our algorithm, we have

$$r_{ij,t} \lambda_{ij,t} = \tilde{r}_{ij,t} \sqrt{\frac{1}{NM-1}} \underbrace{\sqrt{(1 - \frac{\tilde{r}_{ij,t-1}^2}{\|\tilde{\boldsymbol{r}}_{t-1}\|^2})}}_{=:\mathcal{S}_{ij,t}},$$

where the term $\mathcal{S}_{ij,t}$ becomes larger than in the previous step when $r_{ij,t-1}$ is relatively smaller than the other components, and smaller when $r_{ij,t-1}$ is relatively larger. This strategy ensures that $\tilde{r}_{ij,t}^2 \approx \tilde{r}_{ij,t+1}^2$ during training, indicating that the direction of $\boldsymbol{\Lambda}_t \mathbb{A}^\top \tilde{\boldsymbol{r}}_t$ does not change significantly from the initial direction $\tilde{\boldsymbol{r}}_0$.

Assume that we can select a vector $\tilde{\boldsymbol{r}}_0 \notin N(\nabla\theta\boldsymbol{\alpha}(\theta_t))$ for most $t > 0$. In other words, for most of $t$, $\tilde{\boldsymbol{r}}_0$ is outside the null space of $\nabla_\theta\boldsymbol{\alpha}(\theta_t)$. This is a reasonable assumption because in realistic settings, $\nabla_\theta\boldsymbol{\alpha}(\theta_t)$ rarely vanishes, and its null space is a measure-zero set. Based on this assumption, we expect the existence of a constant $c_0 > 0$ such that

$$c_0 := \min_{t \geq 0} \frac{\|\frac{1}{M} \sum_{j=1}^M \nabla_\theta\widehat{\alpha}(\omega_i; \theta_t) \nabla_\theta\boldsymbol{\alpha}(\theta_t) \boldsymbol{v}_0\|_2^2}{\|\frac{1}{M} \sum_{j=1}^M \nabla_\theta\widehat{\alpha}(\omega_i; \theta_t)\|_2^2 \|\nabla_\theta\boldsymbol{\alpha}(\theta_t)\boldsymbol{v}_0\|_2^2},$$

as long as $\|\frac{1}{M} \sum_{j=1}^M \nabla_\theta\widehat{\alpha}(\omega_i; \theta_t)\|_2^2$ does not vanish during training. This implies that under our algorithm, we have

$$q_t \geq c_0 \|\frac{1}{M} \sum_{j=1}^M \nabla_\theta\widehat{\alpha}(\omega_i; \theta_t)\|_2^2,$$

ensuring that $q_t$ has a uniform lower bound, provided that $\left\|\frac{1}{M} \sum_{j=1}^M \nabla_\theta\widehat{\alpha}(\omega_j; \theta_t)\right\|_2^2$ maintains a uniform lower bound—a property that can be demonstrated through various experimental tests in Appendix C.5.

## 5 NUMERICAL EXPERIMENTS

In this section, we present the experimental results of applying the preconditioning technique and our adaptive weight method, as proposed in Section 4, to the ULGNet method. Specifically, we focus on the 1D Helmholtz equation, a paradigm example of a linear elliptic equation, to evaluate the performance of these techniques.

| Trial | Description | Relative L2 Error |
|-------|-------------|-------------------|
| Trial A | ULGNET (baseline) | $2.767 \times 10^{-4}$ |
| Trial B | Preconditioning Type 1 | $1.336 \times 10^{-4}$ |
| Trial C | Preconditioning Type 2 | $1.325 \times 10^{-4}$ |
| Trial D | Type 1 + Adaptive Weight | $8.371 \times 10^{-5}$ |
| Trial E | Type 2 + Adaptive Weight | $\mathbf{7.098 \times 10^{-5}}$ |

Table 1: This table presents the performance of relative $L^2$ errors using different optimization methods. Type A represents the baseline, ULGNet (Choi et al., 2023). In Types B through E, Types 1 and 2 denote different preconditioning methods described in Appendix C.4. The term 'Adaptive Weight' in Types D and E refers to the adaptive weight algorithm outlined in Section 4.

In Table 1, Type A represents the baseline ULGNet (Choi et al., 2023), while Types B through E correspond to various preconditioning methods described in Appendix C.4. The terms 'Type 1' and 'Type 2' denote these methods, with 'Adaptive Weight' in Types D and E referring to the adaptive

weight algorithm outlined in Section 4. Our experiments demonstrate how different preconditioning techniques, both with and without adaptive weights, influence the convergence of the training process. As predicted by our theoretical findings, the trials incorporating both preconditioning and the adaptive weight method (Trials D and E) achieve the best performance, showcasing significantly improved convergence. Notably, Trial E attains the lowest relative $L^2$ error, underscoring the effectiveness of combining Preconditioning Type 2 with the adaptive weight method.

A closer examination can be found in Figure 1, which illustrates the behavior of the variational loss over 50,000 training steps. The experiments reveal how different preconditioning techniques, both with and without adaptive weights, affect the convergence and stability of the training process. Our theoretical findings indicate that trials incorporating both preconditioning and the adaptive weight method (Trials D and E) yield the best performance, with significantly improved convergence stability and reduced final loss values. In particular, Trial E attained the lowest relative $L^2$ error, demonstrating the efficacy of combining Preconditioning Type 2 with the adaptive weight method.

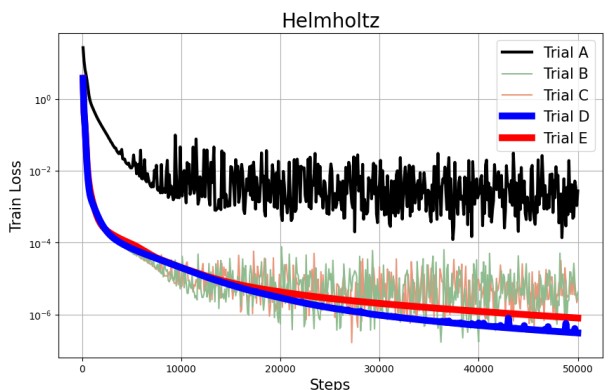

Figure 1: Our theoretical findings indicate that trials incorporating both preconditioning and the adaptive weight method (Trials D and E) yield the best performance, showcasing significantly improved convergence stability and reduced final loss values. Notably, Trial E achieved the lowest relative $L^2$ error, highlighting the effectiveness of combining Preconditioning Type 2 with the adaptive weight method. Furthermore, our proposed approaches in Trials D and E exhibit superior behavior, maintaining stable learning dynamics without oscillations compared to the other trials.

Figure 1 illustrates that our proposed approaches in Trials D and E demonstrate superior performance, attributable to the adaptive weight strategy, which maintains stable learning dynamics without oscillations. Our algorithm plays a crucial role in preventing $q_t$ from vanishing by ensuring that the numerator of the equation equation 12 remains away from zero, thereby contributing to stable training dynamics. In contrast, the other trials exhibit noisy training dynamics and low convergence.

## 6 CONCLUSION AND LIMITATIONS

In this paper, we applied RSC theory to analyze the optimization process in unsupervised operator learning methods utilizing variational loss. Our rigorous analysis demonstrated the convergence of the optimization error, establishing theoretical guarantees for training stability. We highlighted the significance of the condition number of $\boldsymbol{A}$ in influencing convergence rates and showed that preconditioning strategies can substantially enhance training efficiency. Additionally, we addressed the lower bound of $q_t$, proposing an algorithm that adaptively improves this bound without incurring extra computational costs. Our numerical experiments validated the assumptions underlying our theoretical framework and confirmed the effectiveness of preconditioning, revealing significant improvements in training performance and convergence rates.

Despite the promising results, this study has certain limitations. First, while the theoretical analysis is grounded in RSC, the applicability of these results may vary in highly complex or non-standard optimization scenarios. Additionally, the proposed algorithm relies on the assumption that the condition number of $\boldsymbol{A}$ can be effectively managed throughout the training process, which may not hold true in all real-world applications. Future work should explore the robustness of the proposed method in diverse contexts and assess its performance under varying conditions. Moreover, additional empirical studies are needed to further examine the impact of different weight matrices and their configurations on optimization dynamics.

## ETHICS STATEMENT

This research adheres to the ethical standards required for scientific inquiry. We have considered the potential societal impacts of our work and have found no clear negative implications. All experiments were conducted in compliance with relevant laws and ethical guidelines, ensuring the integrity of our findings. We are committed to transparency and reproducibility in our research processes.

## REPRODUCIBILITY

We are committed to ensuring the reproducibility of our research. All experimental procedures, data sources, and algorithms used in this study are clearly documented in the paper. The code and datasets will be made publicly available upon publication, allowing others to validate our findings and build upon our work.

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

## A  RELATED WORKS

**Variational loss in SciML**  Variational loss-based neural networks have emerged as an alternative to traditional PINNs, focusing on enhancing accuracy and stability through the use of variational principles. Prominent examples include VPINN, hp-VPINN, and Galerkin Neural Networks (Kharazmi et al., 2019; **?**; Ainsworth & Dong, 2021), which reformulate the residual minimization problem into a variational form, employing test functions to minimize errors across the entire domain. This approach has demonstrated improved accuracy over standard PINNs, particularly when dealing with weak solutions or complex boundary conditions. In the realm of operator learning, methods such as ULGNET (Choi et al., 2023) and FEONET (Lee et al., 2023) also utilize variational loss forms. Similar to variational neural networks, these operator learning methods incorporate basis functions, enabling them to effectively manage complex domains and challenging boundary conditions. By leveraging these basis functions, these methods provide improved accuracy and stability when solving PDEs in intricate geometries or in the presence of singular perturbations. While this paper focuses on operator networks with variational losses, our approach can also be extended to other operator learning methods or PINNs that employ variational loss functions.

**RSC and NTK**  The NTK theory (Jacot et al., 2018) has been widely utilized to analyze optimization in deep learning (Du et al., 2019; Allen-Zhu et al., 2019; Zou et al., 2020; Wang et al., 2022; 2021a; Gao et al., 2023). While NTK-based approaches offer strong theoretical insights into the convergence of gradient descent, they generally require extremely wide neural networks and depend on the near-initialization regime, which limits their practical applicability. This limitation is particularly evident in the case of PINNs, where NTK theory struggles to address the complexity introduced by PDE residuals (Bonfanti et al., 2024). In contrast, the recently developed RSC theory serves as an alternative tool for analyzing the optimization process of deep learning models. Although RSC has been previously applied to demonstrate geometric convergence in various settings (Wainwright, 2019; Negahban & Wainwright, 2012; Zhang & Cheng, 2015), its application to deep learning was introduced only recently by Banerjee et al. (2022). Building on this work, we extend RSC theory to operator learning methods with variational loss forms, highlighting its potential as a new analytical tool in the field of SciML.

## B  PROOFS

In this section, we provide a series of theoretical proofs for lemmas and main theorems in this work.

### B.1  PROOFS OF LEMMA 3.2 AND 3.3

Before we prove Lemma 3.2 and 3.3, and Theorem 3.4, we first recall some estimates on neural networks established in Banerjee et al. (2022).

**Lemma B.1** (Bound of neural networks $\widehat{\alpha}$). *Under Assumption 1 and 2, for $\theta \in B_{\rho,\rho_1}^{Spec}(\theta_0)$, with probability at least $\left(1 - \frac{2(L+1)}{m}\right)$, we have*

$$\|\widehat{\alpha}_i(\theta;\omega)\|_2 \leq (1 + \phi(0)L)(1 + \rho_1) \tag{13}$$

$$\|\nabla_\theta \widehat{\alpha}_i(\theta;\omega)\|_2 \leq \varrho \tag{14}$$

$$\|\nabla_\theta^2 \widehat{\alpha}_i(\theta;\omega)\|_2 \leq \frac{c_H}{\sqrt{m}}, \tag{15}$$

*where $c_H = \mathcal{O}(poly(L))$ and $\varrho$ is a constant depending on $L$, $\rho_1$ and $\phi$.*

*Proof.*  The proof can be seen in Theorem 4.1 and Lemma 4.1 in Banerjee et al. (2022). $\qquad\square$

Using the bounds of neural networks, we obtain the estimates of the variational loss.

**Lemma B.2** (Bound of variational loss $\mathcal{L}^M$). *Under Assumption 1 and 2, for $\theta \in B_{\rho,\rho_1}^{Spec}(\theta_0)$, the following inequalities hold with probability at least $(1 - \frac{2(L+1)}{m})$:*

$$\mathcal{L}^M(\theta) \leq \sigma_{max}(\boldsymbol{A})^2 c^*. \tag{16}$$

$$\|\nabla_\theta \mathcal{L}^M(\theta)\|_2 \leq \varrho \sigma_{\max}(\boldsymbol{A})\sqrt{2N\mathcal{L}(\theta)} \leq \varrho \sigma_{\max}(\boldsymbol{A})^2 \sqrt{2Nc^*} \tag{17}$$

*where $c^* = N(1 + \phi(0)L)^2(1 + \rho_1)^2 + \frac{1}{M}\sum_{j=1}^{N}\sum_{i=1}^{M}|\alpha_j(\omega_i)^*|^2$.*

*Proof.* We have

$$\mathcal{L}^M(\theta) = \frac{1}{2M}\sum_{i=1}^{M}\|\boldsymbol{A}\widehat{\alpha}(\theta;\omega_i) - \boldsymbol{g}(\omega_i)\|_2^2$$

$$\leq \frac{1}{2M}\sum_{i=1}^{M}\left(\|\boldsymbol{A}\widehat{\alpha}(\theta;\omega_i) - \boldsymbol{A}\widehat{\alpha}_i^*\| + \|\boldsymbol{A}\widehat{\alpha}_i^* - \boldsymbol{g}(\omega_i)\|^2\right)$$

(since $\boldsymbol{A}$ is full-rank, there exists the solution $\widehat{\alpha}_i^*$ s.t. $\boldsymbol{A}\widehat{\alpha}_i^* = \boldsymbol{g}(\omega_i)$ for all $i$)

$$\leq \frac{\sigma_{max}(\boldsymbol{A})^2}{2M}\sum_{i=1}^{M}\left(\|\widehat{\alpha}(\theta;\omega_i) - \widehat{\alpha}_i^*\|_2^2\right)$$

$$\leq \frac{\sigma_{max}(\boldsymbol{A})^2}{M}\sum_{i=1}^{M}\left(\|\widehat{\alpha}(\theta;\omega_i)\|_2^2 + \|\widehat{\alpha}_i^*\|_2^2\right).$$

This, together with (13), implies (16). Then, for $\nabla\mathcal{L}^M$, we obtain that

$$\|\nabla\mathcal{L}^M(\theta)\|_2 \leq \left\|\frac{1}{M}\sum_{i=1}^{M}(\boldsymbol{A}\widehat{\alpha}(\theta;\omega_i) - G(\omega_i))^\top\boldsymbol{A}\nabla_\theta\widehat{\alpha}(\theta;\omega_i)\right\|_2$$

$$\leq \frac{1}{M}\sum_{i=1}^{M}\|(\boldsymbol{A}\widehat{\alpha}(\theta;\omega_i) - G(\omega_i))^\top\boldsymbol{A}\|_2\|\nabla_\theta\widehat{\alpha}(\theta;\omega_i)\|_2$$

$$\leq \frac{\sqrt{N}\varrho\sigma_{\max}(\boldsymbol{A})}{M}\sum_{i=1}^{M}\|(\boldsymbol{A}\widehat{\alpha}(\theta;\omega_i) - G(\omega_i))\|_2$$

$$\leq \sqrt{N}\varrho\sigma_{\max}(\boldsymbol{A})\left(\frac{1}{M}\sum_{i=1}^{M}\|(\boldsymbol{A}\widehat{\alpha}(\theta;\omega_i) - G(\omega_i))\|_2^2\right)^{1/2}$$

$$\leq \varrho\sigma_{\max}(A)\sqrt{2N\mathcal{L}^M(\theta)}$$

$$\leq \varrho\sigma_{\max}(A)^2\sqrt{2Nc^*},$$

which completes the proof. $\qquad\square$

Then, we prove Lemma 3.2 and 3.3 as follows.

**Lemma 3.2.** *Consider a fixed $\theta \in B_{\rho,\rho_1}^{Spec}(\theta_0)$ and $q$ be a fixed positive constant. Under Assumptions 1, 2 and 3, the following inequality holds with probability at least $(1 - \frac{2(L+1)}{m})$: for all $\theta' \in Q_q(\theta) \cap B_{\rho,\rho_1}^{Spec}(\theta_0) \cap B_{\rho_2}^{Euc}(\theta)$*

$$\mathcal{L}^M(\theta') \geq \mathcal{L}^M(\theta) + \langle\theta' - \theta, \nabla_\theta\mathcal{L}^M(\theta)\rangle + \frac{\beta}{2}\|\theta' - \theta\|_2^2,$$

*where $\beta = (\sigma_{\min}(\boldsymbol{A}))^2\left(q - \frac{2\varrho c_H N\rho_2}{\sqrt{m}}\right) - \sigma_{\max}(\boldsymbol{A})^2\frac{c_H\sqrt{2Nc^*}}{\sqrt{m}}$ and $\varrho$, $c_H$, and $c^*$ are given in the previous lemmas.*

*Proof.* We start with the second order Taylor expansion around $\theta$ to obtain

$$\mathcal{L}^M(\theta') = \mathcal{L}^M(\theta) + \langle\theta' - \theta, \nabla_\theta\mathcal{L}(\theta)\rangle + \frac{1}{2}(\theta' - \theta)^\top\frac{\partial^2\mathcal{L}^M(\tilde{\theta})}{\partial\theta^2}(\theta' - \theta) \tag{18}$$

where $\tilde{\theta} = \xi\theta' + (1-\xi)\theta$ for some $\xi \in [0,1]$. For the hessian term in (18), we have

$$\frac{\partial^2 \mathcal{L}^M(\tilde{\theta})}{\partial \theta^2} = \frac{1}{2M} \sum_{i=1}^M \frac{\partial^2}{\partial \theta^2} (\boldsymbol{A}\widehat{\alpha}(\tilde{\theta};\omega_i) - \boldsymbol{g}(\omega_i))^\top (\boldsymbol{A}\widehat{\alpha}(\tilde{\theta};\omega_i) - \boldsymbol{g}(\omega_i))$$

$$= \frac{1}{M} \sum_{i=1}^M \nabla_\theta \widehat{\alpha}(\tilde{\theta};\omega_i) \boldsymbol{A}^\top \boldsymbol{A} \nabla_\theta \widehat{\alpha}(\tilde{\theta};\omega_i)^\top + (\boldsymbol{A}\widehat{\alpha}(\tilde{\theta};\omega_i) - \boldsymbol{g}(\omega_i))^\top \boldsymbol{A} \frac{\partial^2 \widehat{\alpha}}{\partial \theta^2}(\tilde{\theta};\omega_i).$$

Consequently, we obtain that

$$(\theta'-\theta)^\top \frac{\partial^2 \mathcal{L}^M(\tilde{\theta})}{\partial \theta^2}(\theta'-\theta) = \frac{1}{M} \sum_{i=1}^M \|\boldsymbol{A}\nabla_\theta \widehat{\alpha}(\tilde{\theta};\omega_i)(\theta'-\theta)\|_2^2$$

$$+ (\theta'-\theta)^\top (\boldsymbol{A}\widehat{\alpha}(\tilde{\theta};\omega_i) - \boldsymbol{g}(\omega_i))^\top \boldsymbol{A} \frac{\partial^2 \widehat{\alpha}}{\partial \theta^2}(\tilde{\theta};\omega_i)(\theta'-\theta)$$

$$=: I_1 + I_2.$$

For $I_1$,

$$I_1 \geq \frac{\sigma_{\min}(\boldsymbol{A})^2}{M} \sum_{i=1}^M \|\nabla_\theta \widehat{\alpha}(\tilde{\theta};\omega_i)(\theta'-\theta)\|_2^2$$

$$= \frac{\sigma_{\min}(\boldsymbol{A})^2}{M} \sum_{i=1}^M \|(\nabla_\theta \widehat{\alpha}(\theta;\omega_i) + \nabla_\theta \widehat{\alpha}(\tilde{\theta};\omega_i) - \nabla_\theta \widehat{\alpha}(\theta;\omega_i))(\theta'-\theta)\|_2^2$$

$$= \frac{\sigma_{\min}(\boldsymbol{A})^2}{M} \sum_{i=1}^M \Big[ \|\nabla_\theta \widehat{\alpha}(\theta;\omega_i)(\theta'-\theta)\|_2^2 + \|(\nabla_\theta \widehat{\alpha}(\tilde{\theta};\omega_i) - \nabla_\theta \widehat{\alpha}(\theta;\omega_i))(\theta'-\theta)\|_2^2$$

$$+ 2\langle \nabla_\theta \widehat{\alpha}(\theta;\omega_i)(\theta'-\theta), (\nabla_\theta \widehat{\alpha}(\tilde{\theta};\omega_i) - \nabla_\theta \widehat{\alpha}(\theta;\omega_i))(\theta'-\theta)\rangle \Big]$$

$$\geq \frac{\sigma_{\min}(\boldsymbol{A})^2}{M} \sum_{i=1}^M \|\nabla_\theta \widehat{\alpha}(\theta;\omega_i)(\theta'-\theta)\|_2^2$$

$$- \frac{2\sigma_{\min}(\boldsymbol{A})^2}{M} \sum_{i=1}^M \|\nabla_\theta \widehat{\alpha}(\theta;\omega_i)\|_2 \|\nabla_\theta \widehat{\alpha}(\tilde{\theta};\omega_i) - \nabla_\theta \widehat{\alpha}(\theta;\omega_i)\|_2 \|\theta'-\theta\|_2^2$$

where we used the Cauchy-Schwartz inequality in last inequality. We also note that $\tilde{\theta} \in B_{\rho,\rho_1}^{Spec}(\theta_0)$, since $\tilde{\theta}$ is a convex combination of $\theta'$ and $\theta$ in $B_{\rho,\rho_1}^{Spec}(\theta_0)$ which is a convex set. This, together with (14) and (15), implies that

$$I_1 \geq \frac{\sigma_{\min}(\boldsymbol{A})^2}{M} \sum_{i=1}^M \|\nabla_\theta \widehat{\alpha}(\theta;\omega_i)(\theta'-\theta)\|_2^2 - \frac{2\sigma_{\min}(\boldsymbol{A})^2 N}{M} \sum_{i=1}^M \varrho \frac{c_H}{\sqrt{m}} \|\tilde{\theta}-\theta\|_2 \|\theta'-\theta\|_2^2$$

$$\geq \frac{\sigma_{\min}(\boldsymbol{A})^2}{M} \sum_{i=1}^M \|\nabla_\theta \widehat{\alpha}(\theta;\omega_i)(\theta'-\theta)\|_2^2 - \frac{2\sigma_{\min}(\boldsymbol{A})^2 \varrho c_H N}{\sqrt{m}} \|\theta'-\theta\|_2^3$$

where we used the fact $\|\theta'-\theta\|_2 \geq \|\tilde{\theta}-\theta\|_2$ in the last line. Recalling $\theta' \in Q_q(\theta) \cap B_{\rho_2}^{Euc}(\theta)$, we use the definition of the $Q_q(\theta)$ to obtain

$$I_1 \geq \sigma_{\min}(\boldsymbol{A})^2 q \|\theta'-\theta\|_2^2 - \frac{2\sigma_{\min}(\boldsymbol{A})^2 \varrho c_H N \rho_2}{\sqrt{m}} \|\theta'-\theta\|_2^2. \tag{19}$$

Then, for $I_2$,

$$I_2 = \frac{1}{M} \sum_{i=1}^{M} (\theta' - \theta)^\top (\boldsymbol{A}\widehat{\alpha}(\tilde{\theta}; \omega_i) - \boldsymbol{g}(\omega_i))^\top \boldsymbol{A} \frac{\partial^2 \widehat{\alpha}}{\partial \theta^2}(\tilde{\theta}; \omega_i)(\theta' - \theta)$$

$$= \frac{1}{M} \sum_{i=1}^{M} \sum_{j=1}^{N} \{(\boldsymbol{A}\widehat{\alpha}(\tilde{\theta}; \omega_i) - \boldsymbol{g}(\omega_i))^\top \boldsymbol{A}\}_j (\theta' - \theta)^\top \frac{\partial^2 \widehat{\alpha}_j}{\partial \theta^2}(\tilde{\theta}; \omega_i)(\theta' - \theta)$$

$$= \frac{1}{M} \sum_{j=1}^{N} \sum_{i=1}^{M} \{(\boldsymbol{A}\widehat{\alpha}(\tilde{\theta}; \omega_i) - \boldsymbol{g}(\omega_i))^\top \boldsymbol{A}\}_j (\theta' - \theta)^\top \frac{\partial^2 \widehat{\alpha}_j}{\partial \theta^2}(\tilde{\theta}; \omega_i)(\theta' - \theta)$$

For simplicity, we define the following temporary notations:

$$\lambda_{ij} := \{(\boldsymbol{A}\widehat{\alpha}(\tilde{\theta}; \omega_i) - \boldsymbol{g}(\omega_i))^\top \boldsymbol{A}\}_j, \qquad Q_{ij} := (\theta' - \theta)^T \frac{\partial^2 \widehat{\alpha}_j(\tilde{\theta}; \omega_i)}{\partial \theta^2}(\theta' - \theta).$$

From (15), we have

$$|Q_{ij}| \le \|\theta' - \theta\|_2^2 \left\| \frac{\partial^2 \widehat{\alpha}_j(\tilde{\theta}_t; \omega_i)}{\partial \theta^2} \right\|_2 \le \frac{c_H \|\theta' - \theta\|_2^2}{\sqrt{m}}. \tag{20}$$

Then, we use (16) to obtain

$$\frac{1}{M} \sum_{i=1}^{M} \sum_{j=1}^{N} |\lambda_{ij}|^2 = \frac{1}{M} \sum_{i=1}^{M} \sum_{j=1}^{N} |\{(\boldsymbol{A}\widehat{\alpha}(\tilde{\theta}; \omega_i) - \boldsymbol{g}(\omega_i))^\top \boldsymbol{A}\}_j|^2$$

$$= \frac{1}{M} \sum_{i=1}^{M} \|\boldsymbol{A}\widehat{\alpha}(\tilde{\theta}; \omega_i) - \boldsymbol{g}(\omega_i))^\top \boldsymbol{A}\|_2^2 \tag{21}$$

$$\le \frac{(\sigma_{\max}(\boldsymbol{A}))^2}{M} \sum_{i=1}^{M} \|\boldsymbol{A}\widehat{\alpha}(\tilde{\theta}; \omega_i) - \boldsymbol{g}(\omega_i)\|_2^2$$

$$\le 2(\sigma_{\max}(\boldsymbol{A}))^2 \mathcal{L}^M(\theta) \le 2\sigma_{\max}(\boldsymbol{A})^4 c^*.$$

Using the above things, we get

$$I_2 = \frac{1}{M} \sum_{j=1}^{N} \sum_{i=1}^{M} \lambda_{ij} Q_{ij}$$

$$\ge - \left( \frac{1}{M} \sum_{j=1}^{N} \sum_{i=1}^{M} |\lambda_{ij}|^2 \right)^{\frac{1}{2}} \left( \frac{1}{M} \sum_{j=1}^{N} \sum_{i=1}^{M} |Q_{ij}|^2 \right)^{\frac{1}{2}}$$

$$\ge -\sqrt{2}\sigma_{\max}(\boldsymbol{A})^2 \sqrt{c^*} \frac{c_H \sqrt{N} \|\theta' - \theta\|_2^2}{\sqrt{m}}.$$

This, together with (19), gives that

$$(\theta' - \theta)^\top \frac{\partial^2 \mathcal{L}^M(\tilde{\theta})}{\partial \theta^2}(\theta' - \theta)$$

$$\ge \left[ (\sigma_{\min}(\boldsymbol{A}))^2 \left( q - \frac{2\varrho c_H N \rho_2}{\sqrt{m}} \right) - (\sigma_{\max}(\boldsymbol{A}))^2 \frac{c_H \sqrt{2c^* N}}{\sqrt{m}} \right] \|\theta' - \theta\|_2^2,$$

which completes the proof. $\qquad\qquad\square$

**Lemma 3.3.** *Under Assumptions 1, 2 and 3, we have that for all $\theta', \theta \in B_{\rho,\rho_1}^{Spec}(\theta_0)$, with probability at least $(1 - \frac{2(L+1)}{m})$:*

$$\mathcal{L}^M(\theta') \le \mathcal{L}^M(\theta) + \langle \theta' - \theta, \nabla_\theta \mathcal{L}^M(\theta) \rangle + \frac{\gamma}{2} \|\theta' - \theta\|_2^2,$$

*where $\gamma = (\sigma_{\max}(\boldsymbol{A}))^2 \left( \varrho^2 N + \frac{c_H \sqrt{2c^* N}}{\sqrt{m}} \right)$.*

*Proof.* As in the previous lemma, we start with the second order Taylor expansion around $\theta$ to obtain

$$\mathcal{L}^M(\theta') = \mathcal{L}^M(\theta) + \langle \theta' - \theta, \nabla_\theta \mathcal{L}^M(\theta) \rangle + \frac{1}{2}(\theta' - \theta)^\top \frac{\partial^2 \mathcal{L}^M(\tilde{\theta})}{\partial \theta^2}(\theta' - \theta) \tag{22}$$

where $\tilde{\theta} = \xi\theta' + (1-\xi)\theta$ for some $\xi \in [0,1]$. For the hessian term in (22), we have

$$\frac{\partial^2 \mathcal{L}^M(\tilde{\theta})}{\partial \theta^2} = \frac{1}{2M} \sum_{i=1}^M \frac{\partial^2}{\partial \theta^2}(\boldsymbol{A}\widehat{\alpha}(\tilde{\theta};\omega_i) - \boldsymbol{g}(\omega_i))^\top (\boldsymbol{A}\widehat{\alpha}(\tilde{\theta};\omega_i) - \boldsymbol{g}(\omega_i))$$

$$= \frac{1}{M} \sum_{i=1}^M \nabla_\theta \widehat{\alpha}(\tilde{\theta};\omega_i) \boldsymbol{A}^\top \boldsymbol{A} \nabla_\theta \widehat{\alpha}(\tilde{\theta};\omega_i)^\top + (\boldsymbol{A}\widehat{\alpha}(\tilde{\theta};\omega_i) - \boldsymbol{g}(\omega_i))^\top \boldsymbol{A} \frac{\partial^2 \widehat{\alpha}}{\partial \theta^2}(\tilde{\theta};\omega_i).$$

Consequently, we obtain that

$$(\theta' - \theta_t)^\top \frac{\partial^2 \mathcal{L}^M(\tilde{\theta})}{\partial \theta^2}(\theta' - \theta) = \frac{1}{M} \sum_{i=1}^M \|\boldsymbol{A}\nabla_\theta \widehat{\alpha}(\tilde{\theta};\omega_i)(\theta' - \theta)\|_2^2$$

$$+ (\theta' - \theta)^\top (\boldsymbol{A}\widehat{\alpha}(\tilde{\theta};\omega_i) - \boldsymbol{g}(\omega_i))^\top \boldsymbol{A} \frac{\partial^2 \widehat{\alpha}}{\partial \theta^2}(\tilde{\theta};\omega_i)(\theta' - \theta)$$

$$=: I_1 + I_2.$$

For $I_1$,

$$\frac{1}{M} \sum_{i=1}^M \|\boldsymbol{A}\nabla_\theta \widehat{\alpha}(\tilde{\theta};\omega_i)(\theta' - \theta)\|_2^2 \leq \frac{\sigma_{\max}(\boldsymbol{A})^2}{M} \sum_{i=1}^M \|\nabla_\theta \widehat{\alpha}(\tilde{\theta};\omega_i)\|_2^2 \|\theta' - \theta\|_2^2 \tag{23}$$

$$\leq \sigma_{\max}(\boldsymbol{A})^2 \varrho^2 N \|\theta' - \theta\|_2^2. \tag{24}$$

Then, for $I_2$,

$$I_2 = \frac{1}{M} \sum_{i=1}^M \sum_{j=1}^N (\theta' - \theta)^\top (\boldsymbol{A}\widehat{\alpha}(\tilde{\theta};\omega_i) - \boldsymbol{g}(\omega_i))^\top \boldsymbol{A} \frac{\partial^2 \widehat{\alpha}_j}{\partial \theta^2}(\tilde{\theta};\omega_i)(\theta' - \theta)$$

$$= \frac{1}{M} \sum_{i=1}^M \sum_{j=1}^N \{(\boldsymbol{A}\widehat{\alpha}(\tilde{\theta};\omega_i) - \boldsymbol{g}(\omega_i))^\top \boldsymbol{A}\}_j (\theta' - \theta)^\top \frac{\partial^2 \widehat{\alpha}_j}{\partial \theta^2}(\tilde{\theta};\omega_i)(\theta' - \theta).$$

We recall the temporary values used in the proof of Lemma 3.2:

$$\lambda_{ij} := \{(\boldsymbol{A}\widehat{\alpha}(\tilde{\theta};\omega_i) - \boldsymbol{g}(\omega_i))^\top \boldsymbol{A}\}_j, \qquad Q_{ij} := (\theta' - \theta)^T \frac{\partial^2 \widehat{\alpha}_j(\tilde{\theta};\omega_i)}{\partial \theta^2}(\theta' - \theta).$$

By using (20) and (21), we have

$$I_2 = \frac{1}{M} \sum_{j=1}^N \sum_{i=1}^M \lambda_{ij} Q_{ij}$$

$$\leq \left(\frac{1}{M} \sum_{j=1}^N \sum_{i=1}^M |\lambda_{ij}|^2\right)^{\frac{1}{2}} \left(\frac{1}{M} \sum_{j=1}^N \sum_{i=1}^M |Q_{ij}|^2\right)^{\frac{1}{2}} \tag{25}$$

$$\leq \sigma_{\max}(\boldsymbol{A})^2 \sqrt{2c^*} \frac{c_H \sqrt{N} \|\theta' - \theta\|_2^2}{\sqrt{m}}.$$

This, together with equation **??**, gives

$$(\theta' - \theta_t)^T \frac{\partial^2 \mathcal{L}(\tilde{\theta}_t)}{\partial \theta^2}(\theta' - \theta_t) \leq (\sigma_{\max}(\boldsymbol{A}))^2 \left(\varrho^2 N + \frac{c_H \sqrt{2c^* N}}{\sqrt{m}}\right) \|\theta' - \theta\|_2^2$$

$\square$

## B.2 Proof of Theorem 3.4

**Theorem 3.4** (Optimization of the variation loss). *Let $\{\theta_t\}$ denote the sequence of model parameters generated by GD with the stepsize $\eta_t = \frac{\omega_t}{\gamma} \leq \frac{2}{\gamma}$, and we define*

$$q_t = \frac{\sum_{i=1}^{M} \|\nabla_\theta \widehat{\alpha}(\omega_i; \theta_t) \nabla_\theta \mathcal{L}^M(\theta_t)\|_2^2}{M \|\nabla_\theta \mathcal{L}^M(\theta_t)\|_2^2}, \qquad B_t := Q_{q_t}(\theta_t) \cap B_{\rho,\rho_1}^{Spec}(\theta_0) \cap B_{\rho_2}^{Euc}(\theta_t),$$

$$\theta^* \in arginf_{\theta \in B_{\rho,\rho_1}^{Spec}(\theta_0)} \mathcal{L}(\theta), \ \ \bar{\theta}_{t+1} \in arginf_{\theta \in B_t} \mathcal{L}(\theta) \ \ and \ \ \delta_t := \frac{\mathcal{L}(\bar{\theta}_{t+1}) - \mathcal{L}(\theta^*)}{\mathcal{L}(\theta_t) - \mathcal{L}(\theta^*)}.$$

*Under Assumptions 1, 2 and 3, we further assume that for each iteration $t$, the followings holds:*

$$\text{(A1)} \quad \theta_{t+1} \in B_{\rho,\rho_1}^{Spec}(\theta_0) \cap B_{\rho_2}^{Euc}(\theta_t), \ and \qquad \text{(A2)} \qquad q_t > \frac{2\varrho c_H N \rho_2}{\sqrt{m}} + \kappa(A)^2 \frac{c_H \sqrt{2Nc^*}}{\sqrt{m}}. \tag{26}$$

*Then, we have $\delta_t \in [0, 1)$, and the following inequality holds with probability at least $(1 - \frac{2(L+1)}{m})$:*

$$\mathcal{L}^M(\theta_{t+1}) - \mathcal{L}^M(\theta^*) \leq (1 - r_t \omega_t (2 - \omega_t)(1 - \delta_t)) \left(\mathcal{L}^M(\theta_t) - \mathcal{L}^M(\theta^*)\right)$$

*where $r_t$ is given by*

$$r_t = \frac{(\kappa(A))^{-2} \left(q_t - 2\varrho N \frac{c_H \rho_2}{\sqrt{m}}\right) - \frac{c_H \sqrt{2Nc^*}}{\sqrt{m}}}{\varrho^2 N + \frac{c_H \sqrt{2c^* N}}{\sqrt{m}}} > 0,$$

*and $\varrho$, $c_H$, and $c^*$ are given as in Lemma 3.2 and 3.3.*

*Proof.* Since $\theta^* \in arginf_{\theta \in B_{\rho,\rho_1}^{Spec}(\theta_0)} \mathcal{L}(\theta)$ and $\bar{\theta}_{t+1}, \theta_t \in B_{\rho,\rho_1}^{Spec}$, we have

$$\mathcal{L}^M(\theta^*) \leq \mathcal{L}^M(\bar{\theta}_{t+1}) \qquad \text{and} \qquad \mathcal{L}^M(\theta^*) \leq \mathcal{L}^M(\theta_t), \tag{27}$$

which gives $\delta_t \geq 0$. To obtain $\delta_t < 1$, we use Lemma 3.3 and the definition of Gradient Descent to obtain

$$\mathcal{L}^M(\theta_{t+1}) \leq \mathcal{L}^M(\theta) + \langle \theta_{t+1} - \theta_t, \nabla_\theta \mathcal{L}^M(\theta_t) \rangle + \frac{\gamma}{2} \|\theta_{t+1} - \theta_t\|_2^2$$

$$= \mathcal{L}^M(\theta_t) - \eta_t \|\nabla_\theta \mathcal{L}^M(\theta_t)\|_2^2 + \frac{\gamma \eta_t^2}{2} \|\nabla_\theta \mathcal{L}^M(\theta_t)\|_2^2 \tag{28}$$

$$= \mathcal{L}^M(\theta_t) - \eta_t \left(1 - \frac{\gamma \eta_t}{2}\right) \|\nabla_\theta \mathcal{L}^M(\theta_t)\|_2^2.$$

Also, we have from the definition of $Q_{q_t}(\theta_t)$,

$$\theta_{t+1} \in Q_{q_t}(\theta_t) \qquad \text{and hence} \qquad \theta_{t+1} \in B_t.$$

This, together with (27) and (28), gives

$$\mathcal{L}^M(\theta^*) \leq \mathcal{L}^M(\bar{\theta}_{t+1}) \leq \mathcal{L}^M(\theta_{t+1}) \leq \mathcal{L}^M(\theta_t), \tag{29}$$

for $\eta_t \leq \frac{2}{\gamma}$. Thus, we have $\delta_t \in [0, 1)$. To complete the proof, we use Lemma 3.2 to obtain that for any $\theta' \in B_t$

$$\mathcal{L}^M(\theta') \geq \mathcal{L}^M(\theta_t) + \langle \theta' - \theta_t, \nabla_\theta \mathcal{L}^M(\theta_t) \rangle + \frac{\beta}{2} \|\theta' - \theta_t\|_2^2$$

$$\geq \min_{\theta' \in B_t} \mathcal{L}^M(\theta') + \langle \theta' - \theta_t, \nabla_\theta \mathcal{L}^M(\theta_t) \rangle + \frac{\beta_t}{2} \|\theta' - \theta_t\|_2^2$$

$$= \mathcal{L}^M(\theta_t) - \frac{1}{2\beta_t} \|\nabla_\theta \mathcal{L}^M(\theta_t)\|_2^2,$$

where $\beta_t = (\sigma_{\min}(\boldsymbol{A}))^2 \left(q_t - \frac{2\varrho c_H \rho_2}{\sqrt{m}}\right) - \sigma_{\max}(\boldsymbol{A})^2 \frac{c_H \sqrt{2Nc^*}}{\sqrt{m}}$. Consequently, this, together with (28) gives

$$
\begin{aligned}
\mathcal{L}^M(\theta_{t+1}) - \mathcal{L}^M(\bar{\theta}_{t+1}) &\leq \mathcal{L}^M(\theta_t) - \mathcal{L}^M(\bar{\theta}_{t+1}) - \eta_t \left(1 - \frac{\gamma\eta_t}{2}\right) \|\nabla_\theta \mathcal{L}^M(\theta_t)\|_2^2 \\
&\leq \mathcal{L}^M(\theta_t) - \mathcal{L}^M(\bar{\theta}_{t+1}) - \eta_t \left(1 - \frac{\gamma\eta_t}{2}\right) 2\beta_t(\mathcal{L}^M(\theta_t) - \mathcal{L}^M(\bar{\theta}_{t+1})) \\
&= \left(1 - 2\beta_t\eta_t \left(1 - \frac{\gamma\eta_t}{2}\right)\right)(\mathcal{L}^M(\theta_t) - \mathcal{L}^M(\bar{\theta}_{t+1})).
\end{aligned}
$$
(30)

Finally, by using (30) and the definition of $\delta_t$, we have

$$
\begin{aligned}
&\mathcal{L}^M(\theta_{t+1}) - \mathcal{L}^M(\theta^*) \\
&= \mathcal{L}^M(\theta_{t+1}) - \mathcal{L}^M(\bar{\theta}_{t+1}) + \mathcal{L}^M(\bar{\theta}_{t+1}) - \mathcal{L}^M(\theta^*) \\
&\leq \left(1 - 2\beta_t\eta_t \left(1 - \frac{\gamma\eta_t}{2}\right)\right)(\mathcal{L}^M(\theta_t) - \mathcal{L}^M(\bar{\theta}_{t+1})) + \mathcal{L}^M(\bar{\theta}_{t+1}) - \mathcal{L}^M(\theta^*) \\
&= \left(1 - 2\beta_t\eta_t \left(1 - \frac{\gamma\eta_t}{2}\right)\right)(\mathcal{L}^M(\theta_t) - \mathcal{L}^M(\theta^*)) + 2\beta_t\eta_t \left(1 - \frac{\gamma\eta_t}{2}\right)(\mathcal{L}^M(\bar{\theta}_{t+1}) - \mathcal{L}^M(\theta^*)) \\
&\leq \left(1 - 2\beta_t\eta_t \left(1 - \frac{\gamma\eta_t}{2}\right)\right)(\mathcal{L}^M(\theta_t) - \mathcal{L}^M(\theta^*)) + 2\beta_t\eta_t\delta_t \left(1 - \frac{\gamma\eta_t}{2}\right)(\mathcal{L}^M(\theta_t) - \mathcal{L}^M(\theta^*)) \\
&= \left(1 - 2(1-\delta_t)\beta_t\eta_t \left(1 - \frac{\gamma\eta_t}{2}\right)\right)(\mathcal{L}^M(\theta_t) - \mathcal{L}^M(\theta^*)),
\end{aligned}
$$

which, together with $\eta_t = \frac{\omega_t}{\gamma}$, completes the proof. $\qquad\square$

### B.3 Relation between $q_t$ and NTK

In this subsection, we wil give the proof of Theorem 3.5. Recalling the definition of $q_t$, we have

$$
\begin{aligned}
q_t &= \frac{\sum_{i=1}^M \|\nabla_\theta \widehat{\alpha}(\omega_i; \theta_t) \nabla_\theta \mathcal{L}^M(\theta_t)^\top\|_2^2}{M \|\nabla_\theta \mathcal{L}^M(\theta_t)^\top\|_2^2} \\
&= \frac{\sum_{i=1}^M \sum_{j=1}^M |\nabla_\theta \widehat{\alpha}_j(\omega_i; \theta_t) \nabla_\theta \mathcal{L}^M(\theta_t)^\top|^2}{M \|\nabla_\theta \mathcal{L}^M(\theta_t)^\top\|_2^2}.
\end{aligned}
$$
(31)

Here, we use the fact

$$
\begin{aligned}
\nabla \mathcal{L}^M(\theta) &= \frac{1}{M} \sum_{i=1}^M (\boldsymbol{A}\widehat{\alpha}(\omega_i; \theta_t) - \boldsymbol{g}_i)^\top \boldsymbol{A}\nabla_\theta \widehat{\alpha}(\omega_i; \theta_t) \\
&= \frac{1}{M} \boldsymbol{r}(\theta_t)^\top \mathbb{A}\nabla_\theta \boldsymbol{\alpha}(\theta_t),
\end{aligned}
$$

with the definitions of $\boldsymbol{r}$, $\mathbb{A}$, and $\boldsymbol{\alpha}$ given in Subsection 3.3, to obtain that

$$
\begin{aligned}
q_t &= \frac{\sum_{i=1}^M \sum_{j=1}^M |\nabla_\theta \widehat{\alpha}_j(\omega_i; \theta_t) \nabla_\theta \boldsymbol{\alpha}(\theta_t)^\top \mathbb{A}^\top \boldsymbol{r}(\theta_t)|^2}{M \|\nabla_\theta \boldsymbol{\alpha}(\theta_t)^\top \mathbb{A}^\top \boldsymbol{r}(\theta_t)\|_2^2} \\
&= \frac{\|\nabla_\theta \boldsymbol{\alpha}(\theta_t) \nabla_\theta \boldsymbol{\alpha}(\theta_t)^\top \mathbb{A}^\top \boldsymbol{r}(\theta_t)\|_2^2}{M \|\nabla_\theta \boldsymbol{\alpha}(\theta_t)^\top \mathbb{A}^\top \boldsymbol{r}(\theta_t)\|_2^2} \\
&= \frac{\|\nabla_\theta \boldsymbol{\alpha}(\theta_t) \nabla_\theta \boldsymbol{\alpha}(\theta_t)^\top \mathbb{A}^\top \boldsymbol{r}(\theta_t)\|_2^2}{M \boldsymbol{r}(\theta_t)^\top \mathbb{A}\nabla_\theta \boldsymbol{\alpha}(\theta_t) \nabla_\theta \boldsymbol{\alpha}(\theta_t)^\top \mathbb{A}^\top \boldsymbol{r}(\theta_t)} \\
&= \frac{\|\boldsymbol{K}(\theta_t)\mathbb{A}^\top \boldsymbol{r}(\theta_t)\|_2^2}{M \boldsymbol{r}(\theta_t)^\top \mathbb{A}\boldsymbol{K}(\theta_t)\mathbb{A}^\top \boldsymbol{r}(\theta_t)}.
\end{aligned}
$$

This completes the proof.

## C    MORE DETAILS ON NUMERICAL EXPERIMENTS

### C.1    EQUATIONS

In this paper, we consider two linear elliptic equations: a 1D Helmholtz equation and a convection-diffusion problem. Specifically, the equations are given as follows.

- Helmholtz equation reads as

$$
\begin{aligned}
-u_{xx} + ku &= g(\boldsymbol{x}) \text{ in } D,\\
u(\boldsymbol{x}) &= 0 \text{ on } \partial D,
\end{aligned}
\tag{32}
$$

  where we use $k = \frac{7}{2}$ in this paper.

- Convection-diffusion equation reads as

$$
\begin{aligned}
-u_{xx} + \nu u_x &= g(\boldsymbol{x}) \text{ in } D,\\
u(\boldsymbol{x}) &= 0 \text{ on } \partial D,
\end{aligned}
\tag{33}
$$

  where we use $\nu = 1$ in this work.

### C.2    GENERATION OF SAMPLE DATA FOR TRAINING

To generate sample data for training, we use the following $M$ forcing term given as:

$$
g_i = g(\boldsymbol{x}, \omega_i) := \omega_1 \sin(\omega_3(\boldsymbol{x}) + \omega_2 \cos(\omega_4),
$$

where $\omega_1, \omega_2$ and $\omega_3, \omega_4$ are drawn from a uniform distribution on $[3, 5]$ and $[0, 2\pi]$.

### C.3    DETAILS ON EXPERIMENTAL SET-UP

The networks used for the experiments are given as follows.

**Helmholtz Equation**

- The number of hidden layers $L$: 1
- The number of width at each layer $m$: 100
- The number of training samples $M$: 1,000
- The number of basis functions $N$: 30
- Activation function $\phi$: Relu
- Learning rate: 0.001
- Optimizer: Adam
- The number of training steps: 100,000

**Convection-Diffusion Equation**

- The number of hidden layers $L$: 1
- The number of width at each layer $m$: 100
- The number of training samples $M$: 1,000
- The number of basis functions $N$: 30
- Activation function $\phi$: Relu
- Learning rate: 0.001
- Optimizer: Adam
- The number of training steps: 100,000

In our experiments, we applied two types of preconditioning techniques as follows:

- **Type 1:** For this approach, we selected the preconditioner $\mathbf{P}$ as the exact inverse of the matrix $\mathbf{A}$. This method fully compensates for the condition number of $\mathbf{A}$.
- **Type 2:** In this case, we chose $\mathbf{P}$ as a diagonal matrix, where each diagonal entry is the inverse of the corresponding diagonal element of $\mathbf{A}$. While less accurate than Type 1, this method reduces the computational cost associated with the inverse of the full matrix $\mathbf{A}$, providing a balance between efficiency and accuracy.

## C.5    ADDITIONAL EXPERIMENTAL RESULTS

In this subsection, we provide additional figures related to the Helmholtz equation experiments discussed in Section 5. Furthermore, we present the results of applying the same preconditioning and adaptive weight methods to the convection-diffusion equation. Finally, we provide experimental results on the behavior of $q_t$ and the boundedness of the average gradients during the training process. As discussed in Section 3.3, maintaining a uniform lower bound for $q_t$ is crucial for ensuring geometric convergence.

**Helmholtz equation**    We present additional figures related to the Helmholtz equation experiments discussed in Section 5

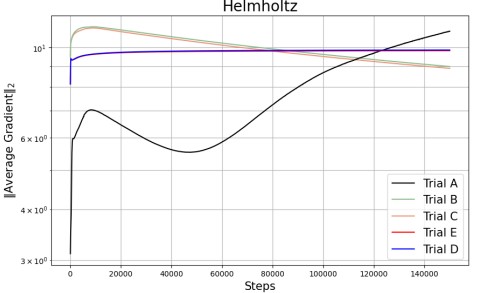

Figure 2: Norm of the average gradient during training for 1D Helmholtz equation.

Figure 3: Loss behavior for 1D Helmholtz equation from different trials.

Figure 4: Behavior of $q_t$ during training for 1D Helmholtz equation.

| Trial | Relative L2 Error |
|-------|-------------------|
| Trial A | $2.767 \times 10^{-4}$ |
| Trial B | $1.336 \times 10^{-4}$ |
| Trial C | $1.325 \times 10^{-4}$ |
| Trial D | $8.371 \times 10^{-5}$ |
| Trial E | $\mathbf{7.098 \times 10^{-5}}$ |

Figure 5: Relative L2 errors for Helmholtz trials.

**Convection Diffusion equation**    We also present experimental results for the convection-diffusion equation. Each type of trial corresponds to those described in Section 5.

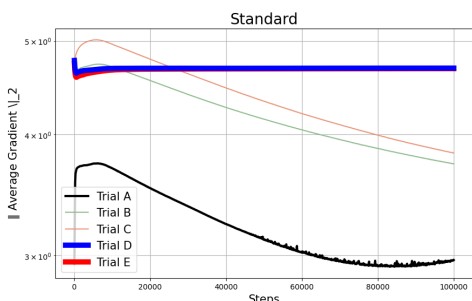

Figure 6: Norm of the average gradient during training for 1D Standard equation.

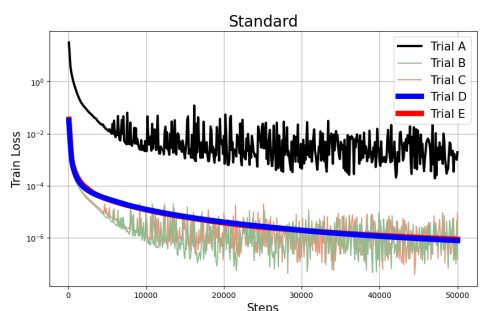

Figure 7: Loss behavior for 1D Standard equation from different trials.

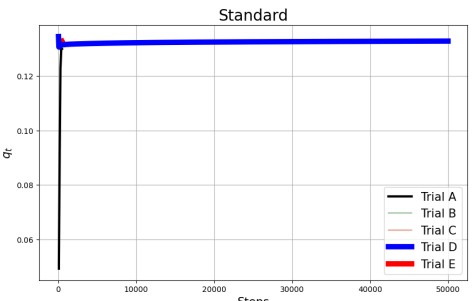

Figure 8: Behavior of $q_t$ during training for 1D Standard equation.

| Trial | Relative L2 Error |
|-------|-------------------|
| Trial A | $1.177 \times 10^{-4}$ |
| Trial B | $7.967 \times 10^{-5}$ |
| Trial C | $8.025 \times 10^{-5}$ |
| Trial D | $\mathbf{7.634 \times 10^{-5}}$ |
| Trial E | $7.860 \times 10^{-5}$ |

Figure 9: Relative L2 errors for 1D Standard trials.

**Behavior of** $q_t$    Figure 10 shows the behavior of $q_t$ over the whole training for both the preconditioned and non-preconditioned cases. Through experiments, we can easily find some cases where $q_t$ vanishes.

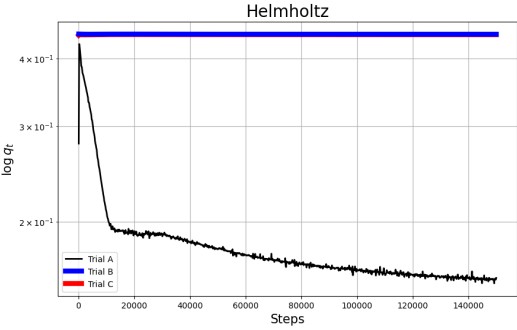

Figure 10: The behavior of $q_t$ during Training. Comparison among Different methods on the 1D Helmholtz equation. Trial A: Original ULGNET; Trial B: ULGNET with Preconditioning Type 1 + Adaptive Weight method; Trial C: ULGNET with Preconditioning Type 2 + Adaptive Weight method.

**Boundedness of the average gradients:**    Figure 11 demonstrates the boundedness of the average gradients $\frac{1}{M} \sum_{j=1}^{M} \nabla_\theta \widehat{\alpha}(\omega_j; \theta_t)$ during the training process. This suggests that the proposed algorithm successfully mitigates the behavior of $q_t$ and ensures that the optimization process proceeds.

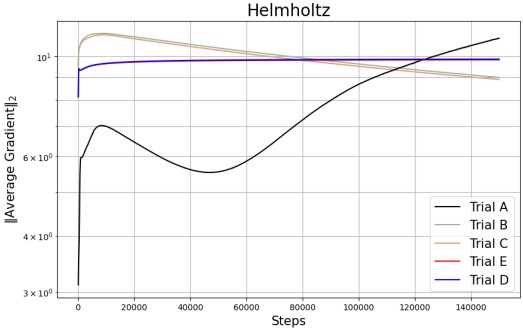

Figure 11: Norm of the average gradient during training for 1D Helmholtz equation. Comparison among Different methods on the 1D Helmholtz equation. Trial A: Original ULGNET; Trial B: ULGNET with Preconditioning Type 1 + Adaptive Weight method; Trial C: ULGNET with Preconditioning Type 2 + Adaptive Weight method.

