# OpenReview forum: "Understanding Optimization of Operator Networks with Variational Loss for Solving PDEs"
_ICLR.cc/2025/Conference — ICLR 2025 Conference Withdrawn Submission_

### Official Review · Reviewer_uKgH · 2024-11-02

**Soundness:** 2
**Presentation:** 2
**Contribution:** 1
**Rating:** 3
**Confidence:** 3

**Summary:**

This paper analyzes the optimization of neural operators for learning solution operators of elliptic PDEs using a variational loss function. The authors use restricted strong convexity (RSC) theory to provide theoretical guarantees for convergence and training stability, and investigate the role of the condition number of the diffusion coefficient $A$ in the optimization.

**Strengths:**

The topic is important and to the best of my knowledge there has not been many works analyzing preconditioning for operator learning techniques. The authors provide some theory and experiments that demonstrate that preconditioning yields better convergence rates and training stability.

**Weaknesses:**

- There is some confusion in the introduction between physics-informed neural networks (for solving known PDEs by minimizing the residual), and neural operator (for learning solution operators without PDE knowledge)
- The paper is not very well structured, and the relevant parts are not very well explained whereas space is wasted on formally introducing neural networks
- The paper is not self-contained and very difficult to understand without reading the paper of Banerjee et al. 2022
- There is some concern about the novelty for the results concerning the condition number of A missing. It is well-known in numerical PDE solvers that smaller condition number improves convergence and preconditioning helps to get smaller condition number
- The main results on convergence rate in Thm 3.4 are not very useful as the convergence rate $r_t$ depends on the quantity $q_t$, which itself depends on the gradient of loss function w.r.t parameters. Hence, one cannot really characterize this as a convergence result given that we have no control on $r_t$.

**Questions:**

- Is there a way to get some expressive bounds on $r_t$? In this current state of Thm 3.4, I can't really say that this characterize the behaviour of the optimization process since all the constants have been hidden in this quantity.
- The approach is only compared to one other approach (ULGNET) but not on standard baseline models (like FNOs or DeepONets)
- Eq. (1): the equation is not self-adjoint due to the term $\nabla u$ contrary to what is stated above
- There should be some regularity condition on the coefficient function (e.g. L^p) to guarantee existence of a solution $u$ in a suitable space.
- Eq. (1), define $d$.
- Paragraph of line 145-155 is very confusing. What's Omega?
- line 179, "classical numerical theory"?
- The beginning of section 3 lack motivation and discussion of previous works.
- Line 334: Section number reference lacking
- line 432-464: all of this is speculative, there is no proof of a uniform lower bound for q_t and this is just a simple consequence of the assumption
- Figure 1: can we not improve the optimization behaviour by tuning the learning rate scheduler?

---

### Official Review · Reviewer_pz5K · 2024-11-03

**Soundness:** 2
**Presentation:** 2
**Contribution:** 2
**Rating:** 1
**Confidence:** 3

**Summary:**

This paper analyzes the optimization of operator networks for solving elliptic PDEs with variational loss functions using the Restricted Strong Convexity framework. It proves convergence, establishes the impact of the condition number on the convergence rate. An adaptive algorithm is proposed to further enhance convergence by adjusting weights in the variational loss function.

**Strengths:**

The paper introduces a new theoretical framework for analyzing the optimization of operator networks with variational losses using the Restricted Strong Convexity theory. This provides a clear image of impact of the condition number on convergence of the algorithm which appears to be in agreement with numerical experiments. The paper further proposes an adaptive preconditioning strategy.

**Weaknesses:**

This paper is extremely difficuly to read, written sloppily, and appears to be significantly underdeveloped. The theory section appears to provide almost no new insights, while the numerics are lacking in both quality and quantity. I expand below:

The effect on the optimization process of the condition number of a linear system for minimizing a linear regression task has been well-known and does not provide any new insights. The proposed adaptive weight algorithm can be interpreted simply as time-dependent pre-conditioning. However, once such a time-dependent pre-conditioning is introduced, theorem 3.4 no longer holds as the algorithm is no longer that of gradient descent, and a new result is required for the updated algorithm - however such an adaptation is not present in the paper.

The numerical section appears to cover only one (there appears to be another one in the appendix, which paints a different picture - showing that adaptive weighting does almost no better than other preconditioning strategies) experiment, reporting only a single relative L2 error (is this over a test set or a single solution?) - making it extremely unclear whether the results are generalizable. The table trial errors do not appear to agree with the graphs provided, however the training curves also do not appear to agree with the original ULGNet of Choi et al. 2023. What is more - the Trial A appears to begin at a much larger loss value, raising the question of whether the loss values reported always correspond to the same loss or different losses - i.e. is it $L^M$ for all runs or $\hat{L}^M$ for some?

I further highlight other issues in detail in the questions section.

**Questions:**

* Abstract appears to refer to A before it is defined. Clasically, A is referred to as the stiffness matrix - so should be replaced.
* Line 99 - you say that input features live in Omega - what is Omega? Is it a set in a vector space? Is it Euclidean? Later on you say it has to be compact. Even later you claim that its L_2 norm must be bounded - does that mean it is a compact subset of euclidean space?
* Line 99 - do you mean to say that $m_0$ is constant $M$? What does 'to be determined later' mean?
* Equation (2) you seem have just dropped the boundary condition in the variational formulation - why is that?
* Assumption 1 appears to have a typo - should be $\beta_\phi$ instead of $\beta_\sigma$.
* Assumption 2 - what happened to the boldening of W and V?
* Definition 3.1 - $\bar{\theta}$ is meant to live in $\mathbb{R}^{p}$, not $\mathbb{R}^{m}$?
* Theorem 3.4: Assumptions A1 and A2 appear to not be discussed at all - what is their significance, why must they hold in practice? Why do you denote the minimization as arginf and not argmin - these are compact sets, and loss is lower bounded?
* Theorem 3.4 $q_t$ is not defined!
* Line 305: "closely related to the convergence rate" - what does that even mean? Either explicitly state what you mean or dont mention.
* Line 334: broken reference
* Line 375 what is \Omega((NM)^2)? \Omega is the feature space?
* Line 380 - you claim that it has been shown that controlling this is challenging in practice - if that is so, please do provide references.
* Section 3 and 4: It appears that things are repeated a lot.
* Algorithm 1 appears to not be written properly, and all actual calcualtions are ommited. What is more, the gradient descent itself is never stated.
* The preconditionings B and C do not appear to be mentioned in the main text at all.
* Why does the relative L2 error for Trial A appear to be 1e-4 while the training loss appears to be 1e-2?
* Why does Trial A begin with a much larger loss value?
* Are the plotted losses always $L^M$ for all runs or $\hat{L}^M$ for some?
* It appears that Trial B-E all operate within the same margin of error, especially for convection diffusion - it seems that adaptive weighting only provides marginal improvement?
* Why is the training loss curves for Helmholtz so different to the original ULGnet paper?

---

### Official Review · Reviewer_vtAK · 2024-11-05

**Soundness:** 2
**Presentation:** 1
**Contribution:** 2
**Rating:** 3
**Confidence:** 4

**Summary:**

The paper uses the recently introduced framework of optimization based on RSC to apply it to a new problem: the training of a neural network that approximates the solution of a specific class of PDE problems. Besides the optimization guarantees, which is the main contribution of the paper, the paper also highlights the role of the preconditioning on the linear system of equations that represents the learning problem, and also proposes an adaptive weight algorithm. Experiments are done to show performance comparisons between different settings of preconditioning and the proposed algorithm.

**Strengths:**

- The paper is well-motivated. It uses a newly introduced optimization framework to provide guarantees on a largely unexplored problem.
- The paper's organization is suitable for the presentation of the results.

**Weaknesses:**

Regarding the theoretical results, I have serious concerns about two specific areas that have not been proved and which imply the proofs are incomplete and therefore not publishable yet.
- In Lemma 3.2, it has not been proved that the set $Q_q\\cap B^{Spec}_{\\rho,\\rho_1}\cap B^{Euc}_{\\rho_2}$ is **non-empty**. I haven’t found the proof for this in the Appendix. If this can’t be proved, then there is no existence of $\\theta’$ and so the equation in Lemma 3.2 is void of meaning, as well as the optimization guarantees in Theorem 3.4 that uses this result—i.e., the main contribution of the paper. The proof is needed urgently for the main theoretical results to make sense.
- There is a big issue in Theorem 3.4. When looking at its proof, it is important to prove that $\\delta_t$ is less than $1$. However, in line 954, when trying to prove this, the authors simply claim that because of the “definition of $Q_{q_t}(\\theta_t)$”, we have that “$\\theta_{t+1}\\in Q_{q_t}(\\theta_t)$”. This claim is not evident and requires a proof—in my opinion, it is a baseless claim. The RSC set $Q_{q_t}(\\theta_t)$ proposed by the authors is different than the RSC set established in Banerjee et al., 2022 (the work that the authors based their results on), and so the results for the RSC set in Banerjee et al., 2022 cannot be applied “immediately” to solve this problem. A proof is needed.

There is a lot of notation problems that may have strong repercussions in the understanding of the paper and possible of the validity of its results. It also shows the paper does not seem to have been proofread. As it is, the paper is not in a suitable state for publication, mainly for the issues I have found below:
- Line 100 mentioned that $M$ is the dimension of the input feature (or data) $\\omega$; however, in equation (6) and later in the same paper $M$ denotes the sample size! This is confusing. Moreover, in Assumption 3, $M$ is used again as saying that it is a constant that simply “exists” for every $\\omega$.
- Another big problem is the use of the symbol $\\omega$. In line 100 it is defined to be an “input feature” which seems to be just data, since it is used in the equation of a neural network. The problem is that then around line 152, it is said to be a member of the “parameter space”, which is a confusing term given that before the paper related $\\omega$ with data. Could the authors make more precise what $\\omega$ represents? I also want to suggest explicitly stating that $\\omega$ is a vector.
- Many typos: for example, delete “through” in line 031; in line 108 it should say “of all layers”; the “(\\mathbf{x})” missing in equation (1); the dimension of $\\bar{\\theta}$ in line 234 should be $p$ instead of $m$; the subscript $2$ must be added to the norm notation in Definition 3.1; remove the $M$ in the denominator from the fraction in lines 1019-1024. The authors should afterwards proofread the whole paper and correct all writing errors.

Finally, another considerable issue is that the simulations are missing a stronger connection with the guarantees established in Theorem 3.4 and the RSC framework.  Theorem 3.4 shows that the convergence rate depends $r_t$, which depends on a different between $q_t$ and a quantity that has $m$ in its denominator. Thus, there seems to be a dependency of the convergence rate on the neural network’s width $m$. Simulations should be done for every of the five trial settings described in Section 5 where now the width values $m$ are changed. The authors should then comment on any observed change in the convergence rate. This would make the simulations explore further the implications of the theoretical derivations.

Other issues:
- The symbol $\\mathbf{A}$ and $q_t$ is introduced in both the abstract and Introduction without any explanation of what they really are. I would recommend to either remove them or give a more anticipatory explanation of what they are.
- Add the Banerjee et al., 2022 citation to the end of the first sentence in line 071.
- Update the Banerjee et al., 2022 citation since I found a published version in ICLR 2023.
- In the contributions, both “training efficiency” and “training performance” are mentioned: what are the differences between those two terms? They must be explained.
- In Section 2.1, specify that $\\phi$ is applied entry-wise when it has a vector as an argument.
- What does “uniformly elliptic” mean in line 121?

**Questions:**

Please, see the Weaknesses question.

---

### Note · Authors · 2024-11-15

I have read and agree with the venue's withdrawal policy on behalf of myself and my co-authors.